# An oomycete plant pathogen reprograms host pre-mRNA splicing to subvert immunity

Jie Huang [1], Lianfeng Gu [2], Ying Zhang[1], Tingxiu Yan[1], Guanghui Kong[1], Liang Kong[1], Baodian Guo[1,3], Min Qiu[1], Yang Wang[1], Maofeng Jing[1], Weiman Xing[3], Wenwu Ye [1,4], Zhe Wu [5], Zhengguang Zhang[1,4], Xiaobo Zheng[1,4], Mark Gijzen [6], Yuanchao Wang [1,4] & Suomeng Dong [1,4]

The process of RNA splicing influences many physiological processes, including plant immunity. However, how plant parasites manipulate host RNA splicing process remains unknown. Here we demonstrate that PsAvr3c, an avirulence effector from oomycete plant pathogen *Phytophthora sojae*, physically binds to and stabilizes soybean serine/lysine/ arginine-rich proteins GmSKRPs. The SKRPs are novel proteins that associate with a complex that contains plant spliceosome components, and are negative regulators of plant immunity. Analysis by RNA-seq data indicates that alternative splicing of pre-mRNAs from 401 soybean genes, including defense-related genes, is altered in GmSKRP1 and PsAvr3c overexpressing lines compared to control plants. Representative splicing events mediated by GmSKRP1 and PsAvr3c are tested by infection assays or by transient expression in soybean plants. Our results show that plant pathogen effectors can reprogram host pre-mRNA splicing to pro- mote disease, and we propose that pathogens evolved such strategies to defeat host immune systems.

[1] Department of Plant Pathology, Nanjing Agricultural University, Nanjing 210095, China. [2] Basic Forestry and Proteomics Research Center, Fujian Provincial Key Laboratory of Haixia Applied Plant Systems Biology, Fujian Agriculture and Forestry University, Fuzhou 350002, China. [3] Shanghai Center for Plant Stress Biology, Shanghai Institutes for Biological Sciences, Chinese Academy of Sciences, Shanghai 201602, China. [4] Key Laboratory of Integrated Management of Crop Diseases and Pests (Ministry of Education), Nanjing 210095, China. [5] Department of Biology, South University of Science and Technology of China, Shenzhen, Guangdong 518055, China. [6] Agriculture and Agri-Food Canada, London, ON N5V 4T3, Canada. Correspondence and requests for materials should be addressed to S.D. (email: smdong@njau.edu.cn)

Plants possess sophisticated defenses and immune systems to protect themselves from pathogens[1]. The initial defense layer is plant cell wall composed of ingredients, including cellulose, pectin, cutin, wax, and callose, which can be reinforced upon pathogen invasion[2]. Plant plasma membrane-based pattern-recognition receptors (PRRs) are the first line of defense and detect pathogen-associated molecular patterns (PAMPs) such as lipopolysaccharides, cutin fragments from fungi, and bacterial flagellin peptides[1,2]. To circumvent PAMP triggered immunity (PTI), pathogens deliver effector proteins into host cells[2,3]. Pathogen effectors are a diverse group of secreted proteins with a broad range of biochemical functions. Effectors often target key components of plant defense pathways and efficiently suppress plant immunity[4]. However, plants have coevolved intracellular NB-LRR receptors known as resistance (R) proteins to recognize certain pathogen effectors, named as avirulence (AVR) effectors, to activate a secondary wave of plant immune responses[1,3]. The R protein-mediated defense response activates dramatic changes in plant hormone-mediated signaling, gene expression, and secondary metabolism, which are powerful determinants of the plant-pathogen interaction[4]. However, the activity of plant R proteins and their signaling network require tight control in order for immune systems to operate properly[5]. The regulation of the immune system during plant-pathogen interactions is an interesting topic of current relevance.

As a key process of RNA metabolism in eukaryotic organisms, RNA splicing processes pre-mRNA transcripts by removing introns from nascent RNA transcripts and joining exons together. Evidence from yeast and mammals suggests splicing happens co-transcriptionally for the majority of genes while post transcriptionally for many individual genes[6]. Common types of alternative splicing (AS) events include intron retention (IR), exon skipping (ES), usage of alternative 5′and 3′splice sites (A5SS and A3SS), and mutually exclusive exons (MXE)[7]. Additionally, the generation of microexons is an AS event that can modulate the function of interaction domains of proteins involved in neurogenesis[8]. Thus, AS increases transcriptome complexity and proteome diversity in multicellular eukaryotes[9]. Recent genome-wide studies indicate that pre-mRNAs from 95% of genes in humans[10], and over 60% of mRNA transcripts from Arabidopsis and soybean undergo AS[11,12]. Pre-mRNA splicing takes place in a large complex called the spliceosome which is made up of several small nuclear RNAs (UsnRNAs) and more than 300 proteins[13,14]. The core of the spliceosome is composed of five small nuclear ribonucleoproteins (U1, U2, U4, U5, and U6 snRNPs) that are highly conserved between metazoans and plants[15]. In addition to snRNPs, many non-snRNPs such as SF1/mBBP and U2AF are also associated in spliceosome operations[16].

Emerging evidence suggests that proper splicing of immunity-related gene transcripts can influence plant disease resistance, but the functional importance of this post-transcriptional modification for plant immunity remains largely unknown[17]. In Arabidopsis, the RPS4 gene confers resistance to the bacterial pathogen Pseudomonas syringae strain Pst DC3000 expressing AvrRps4. The function of RPS4 alternative transcripts, which are generated by intron 2 or 3 retention and splicing of a cryptic intron in exon 3, was tested by stably expressing intron-deprived transgenes in a susceptible Arabidopsis line. Surprisingly, removal of just one intron from the transgene was sufficient to abolish RPS4 function completely. Therefore, resistance to Pst DC3000 requires AS of RPS4[18]. Another example is provided by the RCT1 gene of Medicago truncatula, which confers resistance against multiple races of fungal pathogen Colletotrichum trifoliio[18]. In this case, AS of RCT1 results from the retention of intron 4, instead of intron 2 or intron 3. The alternative isoform ($RCT1_{AT}$) is predicted to encode a truncated protein consisting of the entire

TIR (Toll/interleukin-1 receptor homology), NBS (nucleotide-binding site), and LRR (leucine-rich repeat) domains, but lacking the C-terminal domain of the normal RCT1 protein ($RCT1_{RT}$). Though the expression of RCT1 transcripts was observed to be stable and constitutive, and unaffected by pathogen infection, a certain expression threshold for $RCT1_{AT}$ seemed to be essential for effective resistance[19]. More recent results indicate that Arabidopsis splicing factors SUA and RSN2 are required for proper splicing of the two receptor-like kinases SNC4 and CERK1 and are necessary for PTI response[20]. The gene MOS12 encodes an arginine-rich protein that is homologous to human cyclin L and the mos12-1 mutation resulted in altered splicing patterns of RPS4 and another R gene SNC1[21]. These observations indicate that proper splicing of mRNA transcripts influences immune signaling and disease resistance. However whether plant pathogens interfere with host transcript splicing to suppress immunity remains unclear.

The soil-borne root rot pathogen Phytophthora sojae is one of the most serious threats to global soybean production. This oomycete plant pathogen delivers an arsenal of effector proteins into host cells to enable parasitism[22]. Previously, we identified the AVR gene PsAvr3c from P. sojae by map-based cloning[23]. The predicted PsAvr3c effector protein has a secreted signal peptide and N-terminal RxLR (Arg - any residue - Leu - Arg) motif, which is required for protein translocation into host cells[24]. PsAvr3c has two putative W-motifs, which are likely to be important for maintaining the core structure of the protein. Furthermore, unlike most of the other avirulence effector genes, the PsAvr3c transcript could be detected in all the P. sojae isolates examined so far[23], indicating that PsAvr3c may be required for field fitness. Whether PsAvr3c is required for pathogenesis and what is the mode of action of PsAvr3c remains unclear. In this study, we combined genetics, biochemistry, cell biology and RNA sequencing approaches to address these questions. We show that PsAvr3c binds to GmSKRPs, interactors of spliceosome components. This interaction results in stabilization of GmSKRPs in vivo. In addition, GmSKRPs associate with a complex that contains key splicesome components and more than four hundred genes were differentially spliced in GmSKRPs and PsAvr3c expressing lines. Overall, our current data shed light on a case where a pathogen effector disrupts host RNA splicing machinery to regulate host immunity.

## Results

**PsAvr3c is required for full virulence of *P. sojae*.** To define the expression pattern of PsAvr3c during P. sojae infection, we performed quantitative reverse transcriptase polymerase chain reaction (qRT-PCR) at 0.5,1,2,3,6,24 and 36 hours post inoculation (hpi) on susceptible soybean. The qRT-PCR data demonstrated that the expression of PsAvr3c is induced at early stages of infection and is greatest at 1 hpi, which is consistent with RNA-seq data[25] (Supplementary Fig. 1a, b). In this assay, we used PsXEG1[26] (early expressed glucanase PAMP gene) and PsojNIP[27] (late expressed necrosis-inducing protein-coding gene) as marker genes (Supplementary Fig. 1c, d).

To further investigate the biological role of PsAvr3c, we used the CRISPR/Cas9 genome editing tool to knockout PsAvr3c in P. sojae wild-type strain P6497, which carries PsAvr3c avirulence allele. Two unique sgRNAs were designed to disrupt the PsAvr3c coding region (Supplementary Fig. 2a, b). Protoplast cells of P. sojae were co-transformed with the two sgRNAs and the hSpCas9 expression plasmid. Subsequently, fluorescence selection, gDNA PCR screening, and single colony sequencing identified three independent PsAvr3c knockout mutants (Fig. 1a, Supplementary Fig. 2c, d). Mutants T13 and T61 are

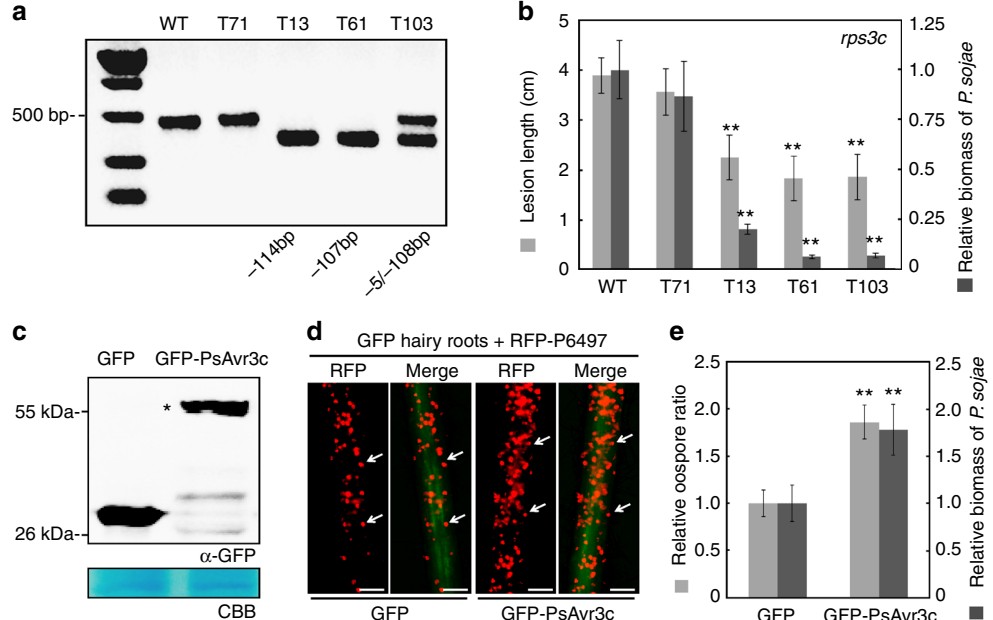

**Fig. 1** Effector gene *PsAvr3c* is required for full virulence of *P. sojae*. **a** Knockout *PsAvr3c* gene by CRISPR/Cas9. The agarose gel shows a PCR result of *PsAvr3c* loci from individual *P. sojae* mutants. Mutants T13 and T61 are homozygous, whereas T103 is a heterozygote. WT represents wild-type *P. sojae* strain P6497. The line T71 is a transformant without *PsAvr3c* editing events. **b** Measurements of lesion lengths and relative biomass on susceptible soybean (*rps3c*). Means and standard errors of lesion lengths from six replicates are shown in light gray (**$P < 0.01$; one-way ANOVA). For relative biomass quantification, the ratio of *P. sojae* DNA compared with soybean DNA was measured at 48 hpi. Means and standard errors of biomass from three replicates are shown in dark gray (**$P < 0.01$; one-way ANOVA). **c** Immuno-blot analysis of GFP-PsAvr3c and GFP in soybean (*rps3c*) hairy roots. The position of GFP-PsAvr3c protein band is indicated by asterisk. The loading control gel was stained with Coomassie brilliant blue (CBB) to visualize protein. **d** Overexpression of GFP-PsAvr3c in soybean (*rps3c*) hairy roots promotes infection. Transformed hairy roots were selected based on the green fluorescence. The hairy roots expressing GFP or GFP-PsAvr3c were inoculated with *P. sojae* strain P6497 expressing RFP. The *P. sojae* oospores were photographed at 48 hpi. The oospores were indicated with arrows. Three independent experiments gave similar results. Scale bars represent 0.25 mm. **e** Statistics analysis and relative biomass quantification of (**d**). Relative oospore ratio is illustrated in light gray. Means and standard errors from different replicates are shown (**$P < 0.01$; one-way ANOVA, $n = 10$). Relative biomass quantification data is shown in dark gray with means and standard errors from six replicates (**$P < 0.01$; one-way ANOVA)

homozygous. A 107 bp fragment was removed between two sgRNA sites in T61, which also produced a frame shift mutation. For T13, two independent 57 bp deletion events occur in each of the sgRNA sites. Mutant T103 is a heterozygote, with 108 and 5 bp deletion in each of the alleles.

Because *P. sojae* strain P6497 possesses two identical copies of *PsAvr3c* in close proximity in genome[23], we examined whether both copies of the gene were affected by the genome editing. Our primers amplified a single product of 459 bp from the wild-type *PsAvr3c* alleles, as expected. However, the PCR products (Fig. 1a) from two independent *PsAvr3c* knockout homozygous mutants show clear single bands (345 and 352 bp, respectively) and no wild-type band can be observed, indicating that both copies were affected by the genome editing. Meanwhile, RT-PCR also validated that *PsAvr3c* mRNA transcripts are impaired in the knockout lines (Supplementary Fig. 2e). An additional transformant, T71, with wild-type *PsAvr3c* allele was selected as control. We found that disrupting *PsAvr3c* in *P. sojae* did not alter its growth rate in vivo compared to strain P6497 (Supplementary Fig. 3a).

To determine the virulence phenotypes of the *PsAvr3c* mutants, we performed an infection assay on *Rps3c* soybean seedlings. The results show that both P6497 and T71 are incompatible on *Rps3c* soybean plants, as expected. In contrast, each of the three *PsAvr3c* knockout mutants can successfully infect *Rps3c* soybeans (Supplementary Fig. 3b, c). Furthermore, the mutants together with wild-type strain are not able to infect soybean plants carrying another resistant gene *Rps3b*

(Supplementary Fig. 3b), indicating that disruption of *PsAvr3c* results specifically in gain of virulence towards *Rps3c* but not *Rps3b* soybeans. To evaluate whether *PsAvr3c* contributes to *P. sojae* virulence in the absence of *Rps3c*, we compared the growth of the *PsAvr3c* mutants and control strain on susceptible soybean plants. The lesion length measurements indicate that the three *PsAvr3c* mutant lines are less virulent compared to control (Fig. 1b). To assess infection more precisely, we used qPCR to measure the relative *P. sojae* biomass (the ratio of *P. sojae* DNA to soybean DNA) in infected tissues. Our data confirm that the virulence of the *PsAvr3c* mutants is significantly impaired on susceptible soybean (Fig. 1b). Taken together, besides its avirulence function, *PsAvr3c* is an essential effector gene that is required for full virulence of *P. sojae*.

To confirm the virulence function of PsAvr3c in planta, a GFP-PsAvr3c fusion protein was transiently expressed in susceptible soybean hairy roots, and then the plants were inoculated with a *P. sojae* RFP strain. The GFP-PsAvr3c recombinant protein of the expected size was detected in transformed soybean hairy roots (Fig. 1c). An inoculation assay indicates that *P. sojae* produces more oospores in GFP-PsAvr3c expressing hairy roots compared to GFP control (Fig. 1d, e). Similarly, the relative *P. sojae* biomass assay confirms that overexpression of GFP-PsAvr3c in soybean hairy roots results in greater growth of the pathogen on the host (Fig. 1e). In summary, these results indicate that *P. sojae* effector PsAvr3c contributes to plant susceptibility.

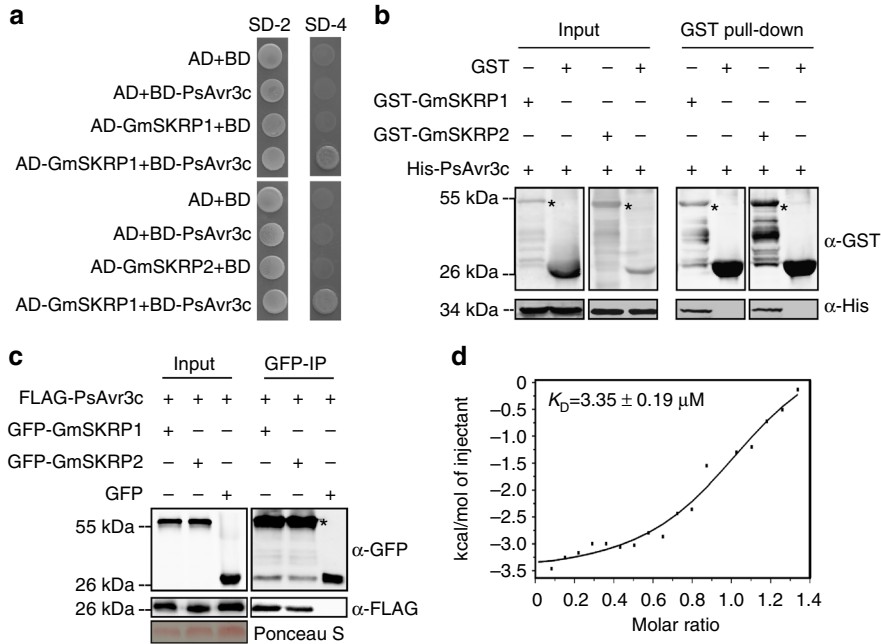

**Fig. 2** PsAvr3c physically interacts with soybean GmSKRP1/2 in vitro and in vivo. **a** PsAvr3c interacts with GmSKRP1/2 in yeast. PsAvr3c and GmSKRP1/2 were cloned into bait plasmid pGBKT7 (BD) and prey plasmid pGADT7 (AD), respectively. Yeast transformants were grown on SD/-Trp/-Leu (SD-2) and selected on SD/-Trp/-Leu/-His/-Ade (SD-4). The plates were photographed 5 days after inoculation. Experiments were repeated three times with similar results. **b** PsAvr3c physically interacts with GmSKRP1/2 in vitro. GST-GmSKRP1/2 or GST bound resins were incubated with *E. coli* supernatant containing His-PsAvr3c. Protein bands of GST-GmSKRP1/2 are marked by asterisks. The presence of His-tagged proteins was detected by western blot using anti-His tag antibody. **c** PsAvr3c interacts with GmSKRP1/2 in vivo. Co-immunoprecipitations (IP) were performed in extracts of *N. benthamiana* leaves expressing both FLAG-PsAvr3c and GFP-GmSKRP1/2. The presence of FLAG proteins was detected by western blot using anti-FLAG antibody. The GFP-GmSKRPs protein bands are indicated by asterisks and the protein loading is indicated by Ponceau stain. **d** Measuring binding affinity between PsAvr3c and GmSKRP1 in vitro. The binding affinity was determined using isothermal titration calorimetry (ITC). The raw titration data were integrated and fitted to a one-site binding model using the MicroCal PEAQ-ITC analysis software. Three original measurement results are 3.38, 3.52 and 3.14 μM, $K_D = 3.35$ μM is mean value of three independent ITC experiments

**PsAvr3c employs a functional nuclear localization signal**. A putative nuclear localization signal (NLS) peptide (KRLNKAWVQYRRKHK) at the C-terminus of PsAvr3c is predicted by motif scan software as shown in Supplementary Fig. 2a. In order to validate whether this predicted NLS is functional, the GFP-PsAvr3c fusion protein was transiently expressed in *N. benthamiana* leaves. Notably, GFP-PsAvr3c protein predominantly accumulated in nucleus (Supplementary Fig. 4a). To further investigate whether the localization of PsAvr3c is dependent on the predicted NLS, we deleted native NLS of PsAvr3c, or substituted native NLS with nls (non-functional NLS) (PKNKRKVEDP)[28] or nuclear exporting signal (NES) (LQLPPLERLTL)[29]. These mutant proteins were excluded from the nucleus (Supplementary Fig. 4a). However, substitution of PsAvr3c native NLS with canonical NLS (QPKKKRKVGG)[30] recovered GFP-PsAvr3c nucleus localization (Supplementary Fig. 4a). The stability and size of each recombinant protein was checked by western blot (Supplementary Fig. 4b).

To test whether the nuclear localization pattern of PsAvr3c is required for PsAvr3c enhanced susceptibility, we inoculated *P. capsici* on the *N. benthamiana* leaves transiently expressing the above PsAvr3c mutants. Only PsAvr3c and its mutants that localize in the nucleus could promote *P. capsici* infection (Supplementary Fig. 4b, c). The relative biomass assay by qRT-PCR confirmed that nuclear localization of PsAvr3c is required for promoting infection (Supplementary Fig. 4d). Collectively, these results suggest that PsAvr3c functions in the nucleus to enhance host susceptibility.

**PsAvr3c interacts with soybean GmSKRP1/2 proteins**. To further elucidate PsAvr3c virulence function, we performed yeast two-hybrid (Y2H) screening to fish for soybean interactors, using PsAvr3c without its secretion signal as bait. Soybean cDNA fragments from two genes (*Glyma.03G234200* and *Glyma.19G231600*) were consistently captured in three independent screens (Supplementary Fig. 5a). We named these genes as GmSKRPs because the proteins are rich in residues of serine, lysine and argine. The coding DNA sequence of *GmSKRP1* and *GmSKRP2* are 558 and 552 bp in length, respectively, encoding predicted proteins with 94% similarity at amino acid sequence level. However, conserved domain prediction by NCBI Conserved Domain Database (CDD) search failed to identify any domain or motif based on GmSKRP1/2 proteins sequence, except for an NLS signal peptide sequence.

To confirm the interactions between PsAvr3c and GmSKRP1/2, four independent assays were carried out. First, we cloned *GmSKRP1/2* full-length cDNA sequences into prey plasmid pGADT7 and validated their interactions with PsAvr3c in yeast (Fig. 2a). Second, we expressed recombinant His-PsAvr3c protein in together with GST-GmSKRP1/2 or GST (as negative control) proteins in *Escherichia coli* (Supplementary Fig. 5b, c), and performed in vitro pull-down assays. As shown in Fig. 2b, the His-PsAvr3c proteins were detected in GST-GmSKRP1/2 but not in GST pull-down products. Third, we performed an in vivo co-immunoprecipitation experiment, and the results confirm the interactions between PsAvr3c and GmSKRP1/2 (Fig. 2c). Finally, isothermal titration calorimetry assay was used to measure the physical binding affinity between PsAvr3c and GmSKRP1 directly. The value of $K_D = 3.35$ μM is a mean value of three

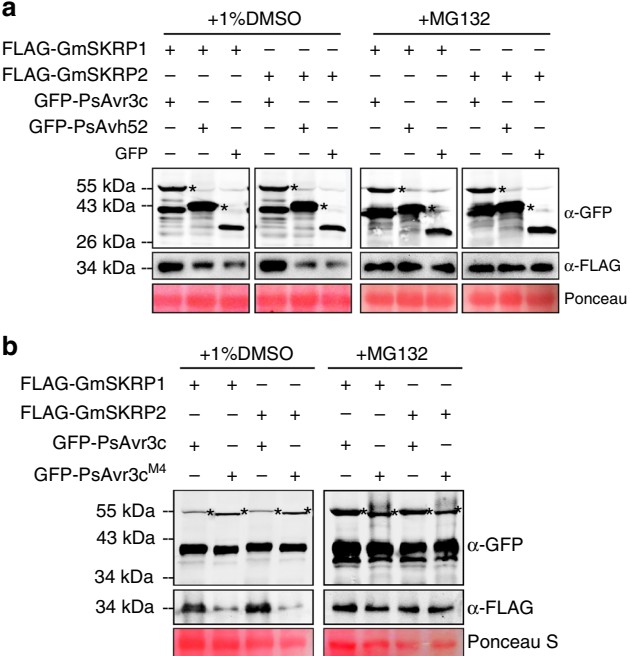

**Fig. 3** PsAvr3c stabilizes GmSKRP1/2 in planta. **a** PsAvr3c stabilizes GmSKRP1/2 proteins in planta. The FLAG-GmSKRP1/2 were co-expressed with GFP-PsAvr3c, GFP-PsAvh52, or GFP, and then infiltrated with 1% DMSO (control) or 50 μM proteasome inhibitor MG132 at 36 hpi. Total proteins were extracted at 48 hpi. PsAvh52 is a nucleus localized effector but does not bind to GmSKRP1/2. Immuno-blotting showed an increased signal from FLAG-GmSKRP1/2 in the presence of GFP-PsAvr3c but not with control protein GFP-PsAvh52 or GFP (left lanes). The degradation of FLAG-GmSKRP1/2 is inhibited by MG132 (right lanes). The position of expected protein bands are indicated by asterisks and protein loading was determined by Ponceau stain. **b** PsAvr3c$^{M4}$ does not stabilize GmSKRP1/2 protein in planta. PsAvr3c$^{M4}$ is a PsAvr3c mutant that fails to bind GmSKRP1/2. Western blotting showing an increased signal from FLAG-GmSKRP1/2 in the presence of GFP-PsAvr3c but not with GFP-PsAvr3c$^{M4}$ (left lanes). The degradation of FLAG-GmSKRP1/2 is inhibited by MG132 (right lanes). The position of expected protein bands are indicated by asterisks and protein loading was determined by Ponceau stain

independent experiments (Fig. 2d). Taken together, these experiments demonstrate that PsAvr3c physically interacts with GmSKRP1/2 in vitro and in vivo.

**PsAvr3c stabilizes GmSKRP proteins in vivo**. To further examine the functions of PsAvr3c effector on GmSKRP1/2 in vivo, we expressed FLAG-GmSKRP1/2 together with PsAvr3c effector in *N. benthamiana*. In this experiment, PsAvh52, another *P. sojae* nuclear localized RxLR effector that does not bind to GmSKRPs, was used as a control (Supplementary Fig. 6). Immuno-blot data indicate that greater amounts of FLAG-GmSKRP1/2 accumulate in vivo in the presence of GFP-PsAvr3c compared to GFP-PsAvh52 and GFP (Fig. 3a). This result suggests that PsAvr3c potentially stabilizes GmSKRPs. Moreover, we found that GmSKRP1/2 in vivo stabilization can be also achieved by infiltrating proteasome inhibitor MG132 (Fig. 3a), suggesting GmSKRP1/2 is consistently degraded by 26S proteasome in vivo. To determine whether greater accumulation of *GmSKRP1/2* might be due to enhanced gene expression, qRT-PCR was used to quantify *GmSKRP1/2* transcript level. As shown in Supplementary Fig. 7, PsAvr3c does not enhance the transcript level of *GmSKRP1/2*. Taken together, the results suggest that PsAvr3c

stabilizes GmSKRP1/2 in vivo by protecting GmSKRPs from proteolytic degradation mediated by the 26S proteasome.

The localization of GmSKRP1/2 proteins and their interaction with PsAvr3c was next examined by confocal microscopy. We observed that RFP-GmSKRP1/2 fusion proteins predominantly accumulated in the nucleoplasm, with a low proportion of nuclear speckle localization (Supplementary Fig. 8a, b). However, co-expression of RFP-GmSKRP1/2 with GFP-PsAvr3c results in the accumulation of GmSKRP1/2 proteins from nucleoplasm to nucleolus (Supplementary Fig. 8c). This effect was never observed in GFP control tests, and GFP-PsAvr3c does not alter RFP protein localization (Supplementary Fig. 8c). Expression of fusion protein expression was verified by western blot (Supplementary Fig. 8d, e).

To further test the biological significance of GmSKRP1/2 and PsAvr3c interaction, we performed PsAvr3c mutant screening. We made a total of five PsAvr3c deletion mutants (PsAvr3c$^{M1}$ - PsAvr3c$^{M5}$). After Y2H screening assay, PsAvr3c$^{M4}$, a 19 amino acid deletion mutant that does not bind GmSKRP1, was identified (Supplementary Fig. 9a). Ectopic expression of PsAvr3c$^{M4}$ in *N. benthamiana* does not stabilize GmSKRPs in vivo (Fig. 3b). Moreover, PsAvr3c$^{M4}$ does not enhance *N. benthamiana* susceptibility (Supplementary Fig. 9b–d), indicating that PsAvr3c binding to GmSKRPs is associated with GmSKRPs stability and PsAvr3c enhanced plant susceptibility. Similarly, PsAvr3c$^{T13}$ and PsAvr3c$^{T103-1}$ proteins produced by *PsAvr3c* CRISPR/Cas9 knockout mutants do not interact with GmSKRP1 (Supplementary Fig. 9a). These findings are consistent with our previous observation that *PsAvr3c* mutants T13 and T103 are less virulent on *rps3c* plants, as shown in Fig. 1b. In summary, the PsAvr3c enhanced susceptibility phenotype appears to be linked with GmSKRPs binding and stabilization.

**GmSKRP1/2 are negative regulators of plant immunity**. Our discovery that GmSKRPs are host targets of the PsAvr3c effector led us to further investigate the potential roles of GmSKRP1/2 in plant defense. We observed that ectopic expression of GFP-GmSKRP1 in *N. benthamiana* promoted *P. capsici* colonization (Supplementary Fig. 10a). Thus, ectopic expression of either the effector PsAvr3c or the target protein GmSKRPs (Supplementary Fig. 10a) in *N. benthamiana* similarly promotes *P. capici* infection. To investigate whether enhanced susceptibility induced by GmSKRP1 and PsAvr3c is synergistic, we co-expressed the proteins in plant cells prior to pathogen challenge. As shown in Supplementary Fig. 10, co-expression of PsAvr3c with GmSKRP1 results in greater susceptibility to *P. capsici* compared to the level of susceptibility induced by expression of either GmSKRP1 or PsAvr3c alone. This synergistic effect was not observed when the PsAvr3c$^{M4}$ mutant was used in place of wild-type PsAvr3c (Supplementary Fig. 10b, c). Therefore, our data suggest that PsAvr3c and GmSKRP1 work through same genetic pathway.

To further examine the roles of GmSKRP1/2 in soybean immunity, we transiently expressed GmSKRP1/2 in susceptible soybean hairy roots. The expression of GFP-GmSKRP1/2 or GFP in transformed hairy roots was detected by immuno-blotting (Supplementary Fig. 11a). The transformed hairy roots were inoculated by *P. sojae* RFP-labeled strain, and after pathogen growth, visualized by fluorescence microscopy. More oospores are visible in the GFP-GmSKRP1/2 hairy roots tissues compared to GFP-transformed control roots, as illustrated in Fig. 4a. The number of *P. sojae* oospores present in GmSKRPs expressing hairy roots is significantly higher than those in control hairy roots (Fig. 4b). Relative biomass assays (Supplementary Fig. 11b) are consistent with the oospore quantifications, and together these

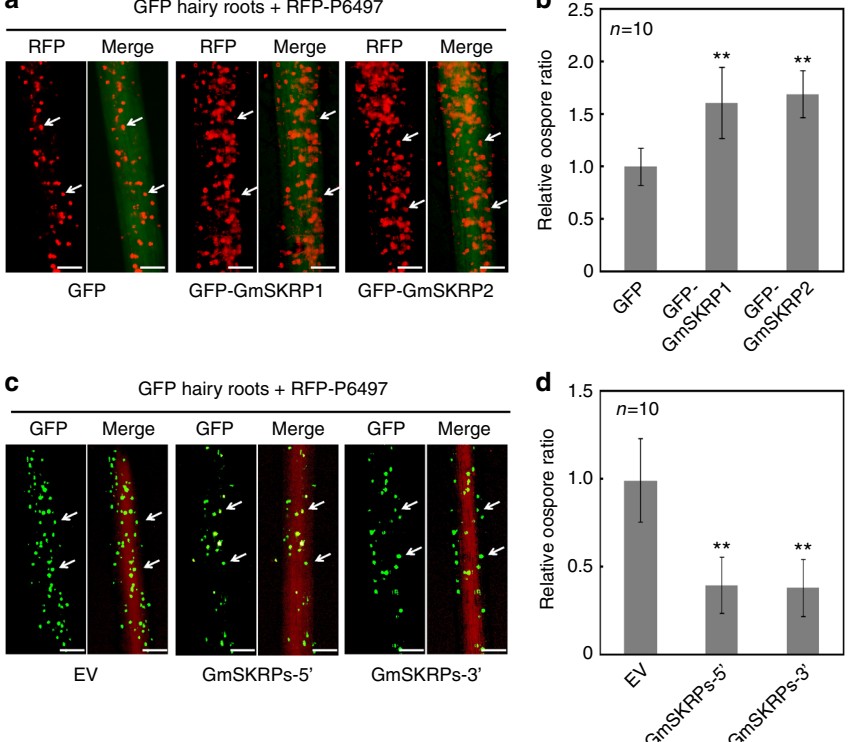

**Fig. 4** GmSKRP1/2 are negative regulators of plant immunity. **a** Overexpression of GFP-GmSKRP1/2 in hairy roots (*rps3c*) promotes *P. sojae* infection. Transformed hairy roots were selected based on the green fluorescence. The fluorescent hairy roots were inoculated with RFP-labeled P6497 strain of *P. sojae*. Images of infected hairy roots were photographed at 48 hpi. The oospores were indicated with arrows. Three independent experiments showed similar results. Scale bars represent 0.25 mm. **b** Statistic analysis of (**a**). Oospore numbers on inoculated hairy roots were measured. Means and standard errors from ten replicates are shown (**$P < 0.01$; one-way ANOVA). Experiments were repeated three times with similar results. **c** Silencing of GFP-GmSKRP1/2 in soybean hairy roots (*rps3c*) enhances soybean resistance. Transformed hairy roots were selected based on the red fluorescence. Silenced hairy roots with RFP marker were inoculated with GFP-labeled P6497 strain of *P. sojae*. GmSKRP1/2 genes were simultaneously silenced by either 5′-end silencing fragment (GmSKRP-5′) or 3′ end silencing fragment (GmSKRP-3′). EV represents empty silencing vector. Infection images were taken at 48 hpi. The oospores were indicated with arrows. Three independent experiments showed similar results. Scale bars represent 0.25 mm. **d** Statistics of (**c**). Oospore numbers were measured in GmSKRPs silencing hairy roots (RFP-GmSKRP-5′ or RFP-GmSKRP-3′) and empty vector (EV) transformed hairy roots. Means and standard errors from ten replicates are shown (**$P < 0.01$; one-way ANOVA). Experiments were repeated three times with similar results

experiments indicate that transient expression of GmSKRP1/2 promotes pathogen growth and infection.

We also conducted transient silencing of GmSKRP1/2 in soybean hairy roots. The *GmSKRP1* and *GmSKRP2* genes share high degree of sequence identity that makes it difficult or impossible to individually silence each gene. Thus, we simultaneously silenced both *GmSKRP* genes in soybean (*rps3c*) hairy roots. Both 5′ and 3′ end consensus sequences from *GmSKRP1/2* were used to make silencing vectors GmSKRPs-5′ and GmSKRPs-3′, respectively. The silencing efficiency assessed by qRT-PCR suggests that *GmSKRP1/2* expression in silenced hairy roots is knocked down by 70% to 80% compared with empty vector (EV) transformed hairy roots (Supplementary Fig. 11c). The *GmSKRP1/2* silenced hairy roots were challenged with GFP-labeled *P. sojae* strain P6497. The number of oospores produced in the *GmSKRPs* silenced hairy roots is severely reduced (Fig. 4c), an observation that is supported by statistical analysis (Fig. 4d). Meanwhile, the relative biomass assay also demonstrates that silencing of *GmSKRPs* in soybean hairy roots enhances resistance to *P. sojae* (Supplementary Fig. 11d). These results indicate that GmSKRP1/2 are negative regulators in soybean immunity.

Given the importance of GmSKRPs to soybean immunity to *P. sojae*, we sought to find orthologous proteins in the model plant *N. benthamiana*, to test whether they may also interact with PsAvr3c and contribute to disease resistance in that species. Searching for *GmSKRP1/2* gene orthologs in *N. benthamiana* genome resulted in the identification of *NbSKRP*. The predicted NbSKRP amino acid sequence shares a high degree of conservation with the GmSKRPs (Supplementary Fig. 12a). Like GmSKRPs, the NbSKRP protein interacts with PsAvr3c, which can be verified by Y2H assay and in vitro pull-down assay (Supplementary Fig. 12b, c). To test *NbSKRP* gene function, both 5′ and 3′ end sequences from *NbSKRP* were used to silence the *NbSKRP* gene in *N. benthamiana* by virus-induced gene silencing (VIGS) approach. The qRT-PCR data indicate that the silencing efficiency induced by NbSKRP-5′ and NbSKRP-3′ is approximately 65 and 75%, respectively (Supplementary Fig. 13a). This level of *NbSKRP* silencing in *N. benthamiana* does not affect plant development and growth (Supplementary Fig. 13b). However, the *NbSKRP*-silenced plants are more resistant to *P. capsici* than the TRV:GFP control plants (Supplementary Fig. 13c–e). These observations also help to explain why ectopic expression of PsAvr3c in *N. benthamiana* leads to enhanced susceptibility to *P. capsici*; GmSKRPs and NbSKRP are orthologous proteins with similar characteristics. Taken together, these experiments demonstrate that soybean and *N. benthamiana* SKRP proteins are negative regulators in plant immunity.

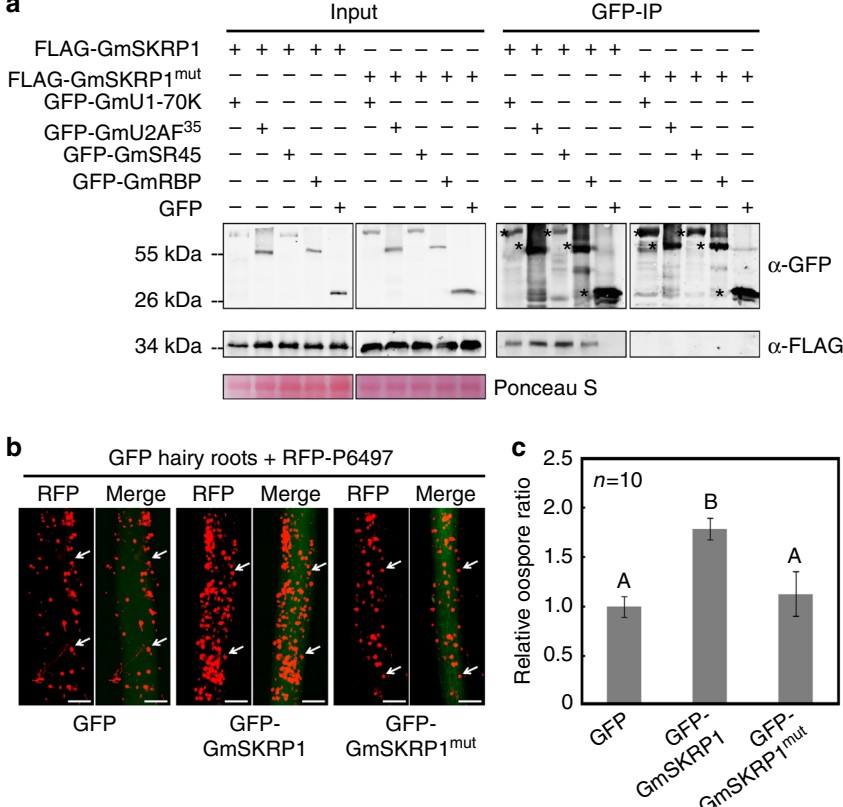

**Fig. 5** Association between GmSKRP1 and soybean spliceosome components is required for susceptibility. **a** GmSKRP1 interacts with complex that contain GmSR45, GmRBP, GmU1-70K, and GmU2AF35 but GmSKRP1mut does not. The co-immunoprecipitation assay demonstrates that GmSKRP1 interacts with GmSR45, GmRBP, GmU1-70K, and GmU2AF35 in planta, but GmSKRP1mut does not. The position of expected protein bands are marked by asterisks and proteins loading was determined by Ponceau stain. **b** Overexpression of GFP-GmSKRP1mut in hairy roots cannot promote *P. sojae* infection. Transformed hairy roots were selected based on the green fluorescence. The fluorescent hairy roots were inoculated with RFP-expressing P6497 strain of *P. sojae*. Images were recorded at 48 hpi. The oospores were indicated with arrows. Three independent experiments showed similar results. Scale bars represent 0.25 mm. **c** Statistic analysis of relative oospore numbers of **b**. Oospore numbers on inoculated hairy roots were measured. Means and standard errors from ten replicates are shown ($P < 0.01$; one-way ANOVA). Experiments were repeated three times with similar results

**GmSKRPs associate with plant spliceosome components in vivo**. To further explore the biological function of GmSKRPs, we expressed FLAG-GmSKRP1 in *N. benthamiana* and harvested the GmSKRP1 binding proteins by co-immunoprecipitation. Since the GmSKRP1/2 and NbSKRP orthologs appear to function similarly in soybean and in *N. benthamiana*, we assumed that the interacting or client proteins for each will likewise be similar. Immunoprecipitation samples were in-gel separated and subjected to LC-MS/MS analyses. The results demonstrate that peptides matching well-studied plant SR-like protein SR45 and a predicted RNA-binding protein (RBP) were consistently recovered (Supplementary Fig. 14). The SR45 and RBP proteins are highly conserved between *N. benthamiana* and soybean, therefore, we cloned full-length cDNAs of soybean *GmSR45* (*Glyma.15G255400*) and *GmRBP* (*Glyma.16G121300*) for further tests. It has been proven that the *Arabidopsis* SR45 protein associates with core spliceosome components[31] such as U1-70K and U2AF35. Therefore, the interactions between GmSKRPs and soybean spliceosomal homologous protein GmU1-70K (*Glyma.01G175100*) and GmU2AF35 (*Glyma.07G159900*), as well as GmSR45 and GmRBP were examined (Fig. 5a, Supplementary Fig. 15a). Although Y2H assay results do not support direct interactions between GmSKRPs and these candidate proteins, with the exception being GmU2AF35 (Supplementary Fig. 15b), the association between GmSKRPs and GmSR45, GmRBP, GmU1-70K, GmU2AF35 can be confirmed by in vivo

co-immunoprecipitation assay (Fig. 5a). These data suggest that GmSKRPs interact with complex that contains several plant spliceosome components in vivo.

To examine whether the association between GmSKRPs and soybean spliceosome components is required for modulation of plant immunity, a GmSKRPs mutant is desired. The GmSKRPs are rich in serine/lysine/arginine residues along the length of the protein sequence, therefore, we synthesized a *GmSKRP1* mutant gene (Supplementary Table 3) with all the arginine and lysine substituted into alanine, except for residues in the NLS motif (Supplementary Fig. 15c). This mutant was named GmSKRP1mut. An in vivo co-immunoprecipitation experiment demonstrates that unlike GmSKRP1, GmSKRP1mut does not bind to GmSR45 and other spliceosome components (Fig. 5a). The GmSKRP1mut displayed attenuated activity in promoting *P. sojae* infection on susceptible soybean plants compared to the wild-type GmSKRP1 (Fig. 5b, c). Thus, the association between GmSKRP1 and soybean spliceosome components is required for GmSKRP1-mediated modulation of plant immunity.

**GmSKRP1 and PsAvr3c regulate pre-mRNA alternative splicing**. Since GmSKRPs associate with a protein complex that contain several key spliceosome components, we assumed that the proteins are likely to be involved in pre-mRNA splicing. To test this hypothesis, RNA-seq data from three independent

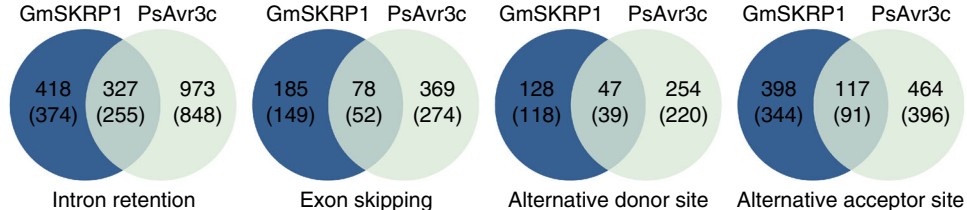

**Fig. 6** RNA-seq analyses of soybean hairy roots ectopically expressing GmSKRP1 and PsAvr3c. Venn diagrams indicating the number of overlapping alternative splicing events and corresponding gene numbers that are significantly alternatively splicing in hairy roots that ectopically express GmSKRP1 or PsAvr3c. The results show that intron retention (327 events) is the most abundant type of event, followed by alternative acceptor site (117 events), exon skipping (78 events), and alternative donor site (47 events). The corresponding gene numbers are shown in parenthesis. Genome coordinates specified individual splicing events being detailed in Supplementary Table 2

soybean GFP-GmSKRP1 overexpression lines, three GFP-PsAvr3c overexpression samples and three GFP control samples were evaluated (Supplementary Fig. 16a). Pairwise comparisons between GFP-GmSKRP1 or GFP-PsAvr3c overexpression lines and GFP were conducted (Supplementary Fig. 16b, c). The RNA-seq data and splicing analyses indicate that a total of 1693 events and 2629 events are differentially spliced in GmSKRPs and PsAvr3c overexpression lines, respectively (Supplementary Fig. 16b, c). A comparison of the results indicates that 569 events are differentially spliced at significant level ($P < 0.05$) in both treatments (Fig. 6, Supplementary Fig. 16d). All the differential alternative splicing events (genes) are listed in Supplementary Table 2. Among four major type of overlapped differential alternative splicing, intron retention (327 events) was the most abundant differential alternative splicing type, followed by alternative acceptor site (117 events), exon skipping (78 events), and alternative donor site (47 events) (Fig. 6). The large numbers of overlapped genes suggest GmSKRP1 and PsAvr3c function under a similar context to regulate splicing, in line with the observed interaction between two proteins. Taken together, these results indicate that GmSKRP1 and PsAvr3c regulate pre-mRNA alternative splicing of a large number of soybean genes.

To verify the RNA-seq analysis of differential alternative splicing, we randomly tested selected AS events by performing semi-quantitative RT-PCR and measuring signal intensities. A total of seven intron retention events and two exon skipping events were examined in soybean hairy roots that ectopically express the five different proteins (Fig. 7). For most intron retention cases, the signal intensity of intron-retained isoforms (upper bands) of most of the transcripts were increased in the presence of ectopically expressed GmSKRP1 or PsAvr3c, whereas, those of intron-spliced isoforms (lower bands) were relatively reduced. The ratio of intron-spliced to intron-retained transcripts was clearly changed. However, the expression of GmSKRP1[mut] and PsAvr3c[M4] mutants in hairy root does not change the ratio significantly. In addition, significant changes of different splicing variants were also observed in two of the exon skipping cases, overexpress GmSKRP1 or PsAvr3c in soybean hairy roots also induces exon skipping of Glyma.19G108300 and Glyma.05G230000 (Fig. 7). Regarding that the limitation of co-amplification PCR may not fully reflect alternative splicing scenario, we further conducted qRT-PCR assay using isoform-specific primers to confirm there are indeed reciprocal changes in the splicing variant (Supplementary Fig. 17). The results showed that expression of spliced isoform decreased while unspliced isoform increased in GmSKRP1 and PsAvr3c overexpressed hairy roots. Therefore, the qRT-PCR data was consistent with the RT-PCR co-amplification data and RNA-seq data. To further test the validity of these results, we examined the alternative splicing events in soybean plants infected with different P. sojae strains

and PsAvr3c knockout mutants. Like the wild-type strain P6497, the two filed isolates FJ8 and AH14 carrying the avirulent allele of PsAvr3c interfere with alternative splicing ratio of the selected genes. However, signals from intron-retained transcripts are clearly reduced or non-detectable during infection by the PsAvr3c knockout mutants T61 and T103. Similarly, exon skipping also disappeared when infected by the PsAvr3c knockout mutants T61 and T103 (Fig. 7). These data confirm that the alternative splicing of host pre-mRNAs is altered in a GmSKRP1 or PsAvr3c dependent manner.

A closer examination of the target genes that were identified from the RNA-seq analysis for alternative splicing events revealed at least two putative defense genes transcripts as intron retention events. To further confirm the RNA-seq and splicing analyses results, we examined the intron splicing ratios (the ratio of spliced RNA over unspliced RNA) of two putative defense genes: a NAC transcription factor (Glyma.02G222300) and a WRKY transcription factor (Glyma.03G220800). Sequence analysis of Glyma.02G222300 shows that it is similar to the Arabidopsis AtNTL9 gene, which encodes a protein suggested to be a bacteria effector target and a positive immune regulator[32] (Supplementary Fig. 18a). Likewise, the soybean WRKY gene Glyma.03G220800 is similar to the Arabidopsis AtWRKY28 gene (Supplementary Fig. 18b), which encodes a key transcription factor involved in SA synthesis[33]. Another related gene AtWRKY23, is involved in nematode infection[34]. An inoculation assay also shown that expressing Glyma.02G222300 and Glyma.03G220800 cDNA sequence in soybean hairy roots are more resistant to P. sojae (Supplementary Fig. 18c–g). The difference in splicing ratio of the two soybean defense-related genes was also detected by qRT-PCR in GmSKRP1 and PsAvr3c overexpression hairy roots, but cannot be detected in the GmSKRP1[mut] and PsAvr3c[M4] overexpression hairy roots (Fig. 8a, b). As a control, intron splicing of a soybean COP9 signalosome complex subunit gene (Glyma.03G016800, Glyma.03G016800 was previously named as Glyma03g01880 and had an intron retention event)[12] was not affected (Fig. 8c). Meanwhile, qRT-PCR data also demonstrated that the transcript level of all three tested genes are not significantly altered among hairy roots expressing GmSKRP1, PsAvr3c and their mutants (Supplementary Fig. 19a–c). It is likely that these abnormal splicing variants may not translate into functional resistance proteins based on previous studies[35,36]. These results demonstrate that ectopic expression of either GmSKRPs or PsAvr3c in soybean hairy roots affects the splicing of soybean genes, including predicted defense-related genes.

In order to determine whether the PsAvr3c effector affects host pre-mRNA splicing during P. sojae infection of soybean plants, we inoculated P6497 wild-type strain together with field isolates FJ8, AH14 and three different PsAvr3c knockout mutants on a susceptible soybean line. Total RNA samples were extracted and qRT-PCR was performed to quantify splicing ratio of selected

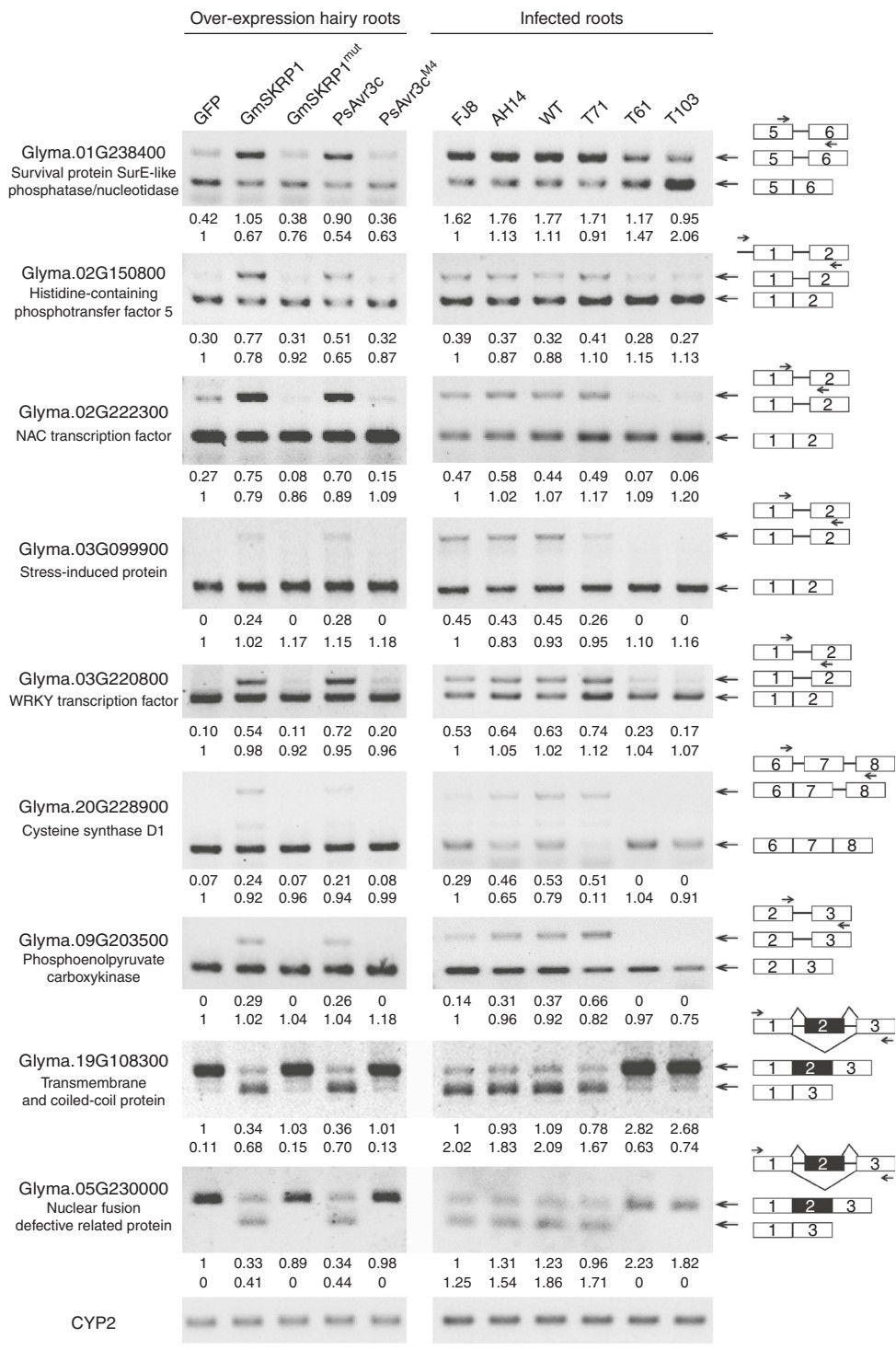

**Fig. 7** Validation of expression and alternative splicing of selected genes. The changes in splice isoforms of nine selected genes was validated by RT-PCR with primers specific to each isoform. An equal amount of cDNA template in each reaction was verified by amplifying soybean actin gene CYP2, and then PCR reaction with primers specific to each AS isoform was conducted (Primers were listed in Supplementary Table 1). The intensities of each product were quantified by ImageJ software, and the relative abundance was calculated by using bottom (or top) bands in the left electrophoresis line as reference (set to 1). Numbers below the PCR bands indicate relative abundances of different isoforms. Schematic diagrams on the right side illustrate the AS isoforms of each gene and the arrows highlight the primers used in RT-PCR. The *P. sojae* strains FJ8 and AH14 are field isolates and carry the avirulent allele of *PsAvr3c*. Other transformants refers to the Fig. 1

genes. As shown in Fig. 8a, b, we verified that the splicing ratios of the soybean NAC and WRKY mRNA transcripts are significantly enhanced in plants infected with the mutant *P. sojae* strains. These data are consistent with the previous observation that splicing ratio is impaired in PsAvr3c expressing soybean hairy roots. In summary (Fig. 8d), our current evidence demonstrates that PsAvr3c effector targets GmSKRPs proteins to reprogram soybean pre-mRNA splicing, and the splicing change of soybean resistance proteins is likely to interfere plant immunity.

## Discussion

Emerging evidence points to a role for RNA splicing as a mechanistic control point for plant development and immunity[17]. However, whether parasites manipulate the plant RNA splicing process to suppress host immune system remains unknown. In the present study, we identified SKRP proteins as pathogen virulence targets and novel regulators in plant pre-mRNA splicing machinery by screening for host clients of the *Phytophthora sojae* core effector PsAvr3c. Genes encoding

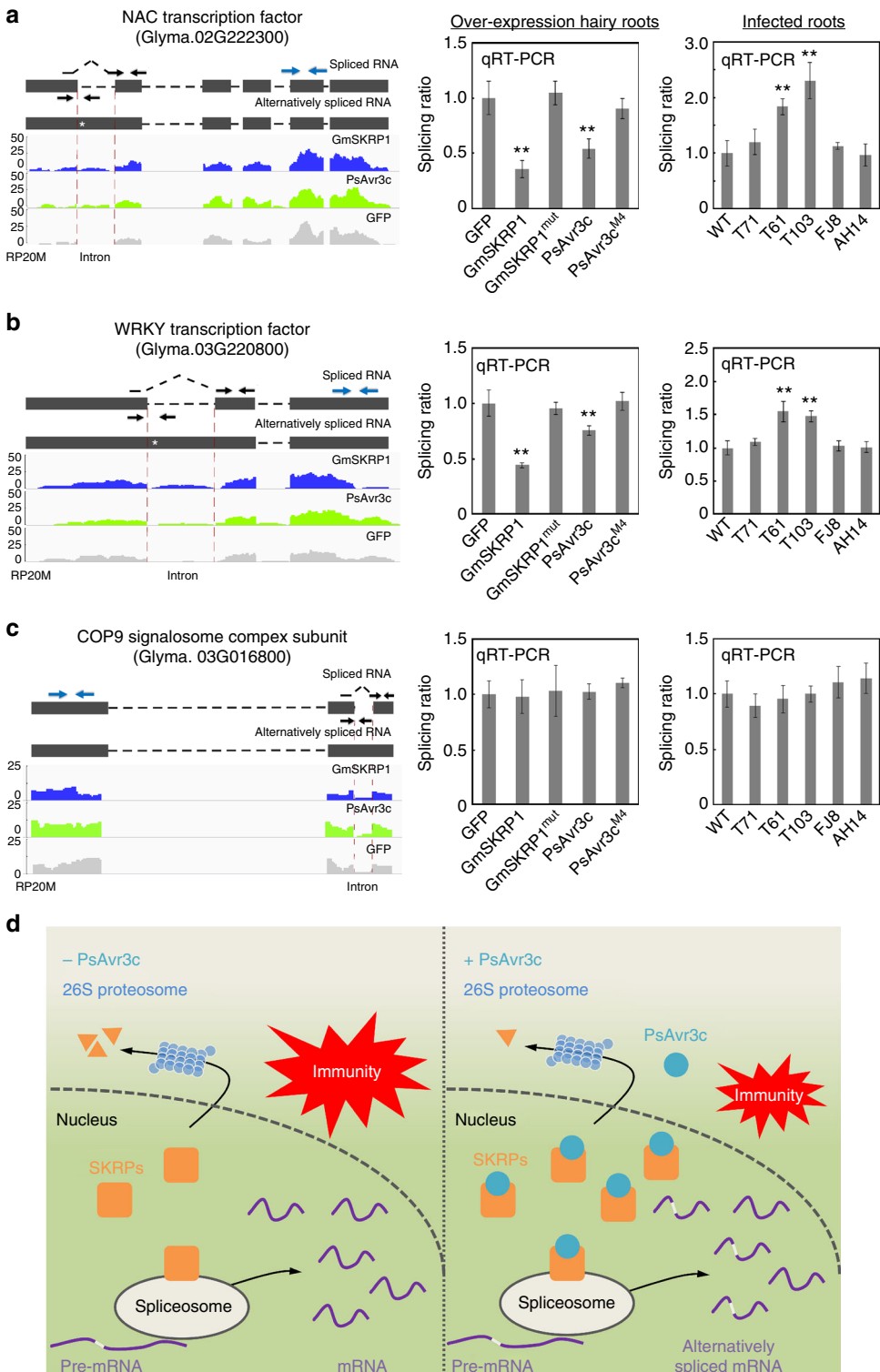

SKRP-like proteins are prevalent in plants (Supplementary Fig. 20a, Supplementary Table 3) and likewise PsAvr3c effector homologs are predicted in several of *Phytophthora* genomes (Supplementary Fig. 20b, Supplementary Table 3). Thus, it is plausible that our findings are more widely relevant and that SKRP-effector interactions occur in other pathosystems. In any case, our results provide new evidence that a pathogen effector interferes with host RNA splicing machinery to reprogram the splicing of pre-mRNA, and we propose that pathogens suppress host immunity at pre-mRNA splicing layer (Fig. 8d).

The conclusion that PsAvr3c subverts host pre-mRNA splicing fits and complements existing theories and models of how pathogen effectors operate. For example, a previous study demonstrated that the bacterial pathogen *Pseudomonas syringe* injects the type III effector HopU1 to ADP-ribosylate plant RNA-binding protein (RBPs) such as GRP7, to suppress plant immunity[37,38]. A KH type RBP was also identified as an important susceptibility factor for colonization of the late blight pathogen *Phytophthora infestans* on potato[39]. More recently, the discovery that a *Phytophthora sojae* effector targets a plant RNA helicase protein to regulate the accumulation of small interfering RNAs and microRNA illustrates how a pathogen can manipulate host small RNA metabolism[40]. These discoveries together with our results present a compelling case that eukaryotic and prokaryotic plant pathogens evolved effector systems to alter host RNA metabolism through a variety of control points and mechanisms.

Specifically, the control point targeted by PsAvr3c is one that is dependent on the host proteins GmSKRP1/2. Although we found that PsAvr3c is able to stabilize GmSKRP1/2 from 26S proteasome-mediated protein degradation, it is not clear how this enhanced stability is achieved. One of the possibilities is that PsAvr3c binding masks the key residues of GmSKRP1/2 that are required for ubiquitination, and therefore perturbs the GmSKRP1/2 as substrates of E3 ubiquitin ligase. This possibility is supported by the observations that GmSKRPs are consistently degraded in vivo in a 26S proteasome dependent manner, and that PsAvr3c prevents GmSKRPs degradation in a manner, which is similar to the proteasome inhibitor MG132. To further test this hypothesis, a GmSKRPs mutant that does not bind PsAvr3c and a crystal structure of PsAvr3c-GmSKRP complex would be informative. Our confocal microscopy observations could offer other clues as to how PsAvr3c stabilizes GmSKRPs. We observed that GmSKRPs mainly localize in the nucleoplasm in the absence of PsAvr3c, whereas they shift into the nucleolus in the presence of PsAvr3c. The GmSKRPs subcellular shift from the nucleoplasm into the nucleolus driven by PsAvr3c may prevent their accessibility or exposure to specific E3 ubiquitin ligases. Furthermore, one could argue that GmSKRPs could be alternatively spliced in the presence of PsAvr3c and generate a more stable isoform. We looked through the RNA-seq data and did not find significant AS events for GmSKRP1/2 in PsAvr3c hairy roots. Meanwhile, we did not observe additional GmSKRPs protein bands in the presence of PsAvr3c from our western blot data (Fig. 3). Therefore, we speculate that GmSKRP1/2 have no detectable AS isoform which could explain the increased stability of GmSKRPs. Other possibilities such as lower turnover rate due to retention of GmSKRP to spliceosome complex may also explain the increased amount of the GmSKRPs. Nevertheless, the mode of action of PsAvr3c effector on GmSKRPs or GmSKRPs-spliceosome complex remains to be investigated.

The development of experimental plant lines that overexpress GmSKRPs provides a useful model to investigate how PsAvr3c induced stabilization of the proteins promotes disease and pathogen growth. The RNA-seq data indicate that over 400 soybean genes differ significantly in terms of AS pattern in either GmSKRPs or PsAvr3c overexpressed lines compared to controls. Clearly, the overexpression of either GmSKRPs or PsAvr3c has profound effects on pre-mRNA splicing outcomes. Analysis of genes that are differentially spliced in the presence of ectopically expressed SKRPs and PsAvr3c suggests that GmSKRPs are regulators of RNA splicing and that PsAvr3c indirectly modulates soybean pre-mRNA splicing process through GmSKRPs. Strikingly, more than half of AS events are intron retention type. It is possible that the reduced splicing ratio might result from stabilization of intron-retained isoforms, rather from a change in real alternative splicing. This led us to think about RNA nonsense-mediated decay (NMD) inhibition, which has been previously reported as an important virulence strategy for plant and animal virus[41,42]. The co-amplification of splicing variants by semi-quantitative RT-PCR demonstrated that the ratio changes between splicing variants is not only due to significant accumulation of intron-retained isoforms, a clear reduction of intron-spliced isoforms are also observed in most of the cases (Fig. 7). Due to limitation of PCR reaction in co-amplification, we further performed RT-qPCR of individual splicing variants from each of the selected gene, and found that there are reciprocal changes in the splicing variant levels (Supplementary Fig. 17). Thus, RNA-seq, co-amplification RT-PCR and RT-qPCR results clearly indicated that most of the splicing variation changes are due to pathogen effector-mediated pre-mRNA alternative splicing. Meanwhile, we indeed observed dramatic accumulation of intron-retained isoform without decreasing of other isoforms in a few cases which fit NMD inhibition scenario. Given that, there are premature stop codons in retained isoform, we cannot rule out the possibility that NMD inhibition may partially involve in GmSKRP1 or PsAvr3c-mediated splicing ratio changes. Beside NMD, some other processes such as transcripts transport and turnover processes may also affect splicing variant ratio change. Whether these processes are also involved in PsAvr3c-mediated host susceptibility remains further investigations.

Further analysis by semi-quantitative RT-PCR and qRT-PCR on selected genes confirmed the splicing pattern in the GmSKRP1

**Fig. 8** Validation of defense-related genes pre-mRNAs splicing ratio by qRT-PCR. **a–c** Selected gene splicing ratio and gene expression was determined by RNA-seq and qRT-PCR analyses. The *GmNAC* and a *GmWRKY* predicted transcription factor genes are subject to AS in the GmSKRP1 and PsAvr3c overexpressing lines, whereas the COP9 signalosome complex subunit gene does not. The left panel shows wiggle plots of RNA-seq data with the schematic gene model, the asterisks indicate the premature stop codon. The black and blue arrows indicate the primers that are used to measure the intron splicing ratio and gene relative expression, respectively. The spliced primers that cross exon-exon junctions are shown with dashed lines over the intron. Splicing ratio was calculated by determining the level of spliced RNA normalized to the level of unspliced RNA. Values on the Y-axis indicate reads per 20 million (RP20M). The right panel shows the selected genes splicing ratio of introns in five different overexpression hairy roots and six different infection roots. Pre-mRNA intron splicing ratio from three replicates are normalized and presented as means ± standard errors (**$P < 0.01$; one-way ANOVA). **d** Schematic representation of *PsAvr3c* virulence function. In absence of PsAvr3c effector (left), GmSKRPs are consistently degraded by 26S proteasome probably for maintaining protein turnover. SKRPs associate with spliceosome components to regulate pre-mRNA splicing. Proper splicing of defense-related genes contribute to plant resistance. In the presence of PsAvr3c effector and the absence of *Rps3c* (right), PsAvr3c associates and stabilizes SKRPs in plant nucleus. Increased level of SKRPs results in large scale of pre-mRNA alternative splicing. In particular, some defense-related genes generate alternative splicing isoforms, thus the PsAvr3c effector reprograms host pre-mRNA splicing to suppress plant immunity

or PsAvr3c overexpressing line but not in the case of ectopic expression of their non-functional mutants. For the putative defense genes *GmNAC* and *GmWRKY* that displayed altered pre-mRNA splicing patterns, we noted that only one of the introns in each of the two genes was retained, whereas the other introns were clearly processed. Among the 401 genes identified, further work is necessary to determine how altered GmSKRPs levels cause such massive but specific changes in mRNA processing. One explanation could be that GmSKRPs are specifically involved in regulating AS of a proportion of the soybean transcriptome. Such a scenario is likely to involve the context specific recognition of targets via other splicing regulators like SR45. The accurate reconstruction of transcript isoforms from short reads (such as provided by the Illumina platform) remains a technical challenge[43], Thus, the identification and quantification of alternative splicing events may not be fully complete or reliable. The actual number of affected splicing events could be different than we identified in this initial analysis. To assess the genes that are subject to AS on the genome-wide level during infection by PsAvr3c effector, an alternative approach would be direct comparison of RNA-seq data from soybean tissues that are infected by wild-type strain and *PsAvr3c* knockout mutant. This approach seems to be more biological appealing, but things such as effector redundancy and reconstruction of AS remain significant challenges. Nonetheless, our finding that host pre-mRNAs are alternatively spliced in the presence of the effector PsAvr3c is certainly consistent with results from the GmSKRP overexpression lines. We hypothesize that SKRPs are maintained at a prescribed level in the plant cell to ensure AS operations are properly controlled. Furthermore, pathogen effectors such as PsAvr3c enhance SKRPs accumulation and perturbs AS. These effects result in altered transcriptional outcomes, at least for a proportion of expressed genes (Fig. 6c).

Non-model systems present challenges but also opportunities for discovery of new functionalities for gene or protein sequences with limited annotation. Previous research in animal and plant systems defined many splicing proteins with a modular structure containing one or two RNA recognition motifs (RRM). The RRM occurs at the N-terminus and provides RNA-binding specificity, and a C-terminal RS domain acts to promote protein–protein interactions[44]. However, GmSKRPs do not have any RRM or any other predicted functional domain, albeit the full-length protein is rich in serine/lysine/arginine residues. Moreover, our LC-MS/MS and in vivo Co-IP data suggest that GmSKRPs associate with SR45 and key spliceosome components, including U1-70K and U2AF35. These data are consistent with previous results showing that SR45 binds to the plant spliceosome subunits U1-70K and U2AF35. Although GmSKRPs do not contain any known domains, we found that the lysine and arginine residues are crucial for protein functionality. We acknowledge that a large number of residues were substituted in our GmSKRP1 mutant, and that additional GmSKRP1 mutant screening will be required to pinpoint key residues or regions that are involved in function.

Although the SKRPs appear to be unrelated to conventional RBPs, the two types of proteins may share certain characteristics. Plant glycine-rich RBP proteins are known for RNA binding. Recently, it was found that the *Arabidopsis* RZ-1B/C proteins interact with SR proteins and bind to RNA. In one study, the authors report that deletion of *RZ-1B/C* is accompanied by large-scale defective gene splicing and changes in gene expression[45]. In another case, taking advantage of RNA immunoprecipitation and sequencing technology, more than 4,000 *Arabidopsis* RNAs that directly or indirectly associate in vivo with the SR45 protein were identified[46]. Our finding that GmSKRPs interact with known spliceosome components helps to further

define plant pre-mRNA processing factors. Whether SKRPs recruit specific substrate RNAs is a relevant question that arises from our work.

Previous studies have shown that *P. sojae* PsAvr3c is a core AVR effector that is expressed during the biotrophic phase of infection, in all the tested isolates[23]. Now, with the advantage of the CRISPR/Cas9 editing tool, we knocked-out *PsAvr3c* and found that this compromised the growth and aggressiveness of the pathogen on susceptible host plants. Furthermore, transient expression of PsAvr3c in soybean and in *N. benthamiana* enhances plant susceptibility to pathogen attack. Together these data suggest that PsAvr3c is a virulence factor that suppresses plant immunity to enable pathogen growth. This is consistent with the hypothesis that effector-triggered susceptibility and effector-triggered immunity are the two sides of effector biology that engage with plant immune systems and result in contrasting disease outcomes.

Although the cognate soybean resistant gene *Rps3c* has not been identified or characterized, we propose that PsAvr3c could be recognized directly by the Rps3c protein, or indirectly through a guardee protein that is part of the Rps3c immune signaling complex. Another possibility is that alternative splicing event(s) caused by PsAvr3c triggers Rps3c-mediated immunity. Predicted genes with homology to PsAvr3c are present in a wide range of *Phytophthora* species, so it is possible that underlying full understanding of PsAvr3c triggered immunity may help in developing successful strategies to manage other *Phytophthora* diseases.

## Methods

**Plant and microbe cultivation**. *N. benthamiana* plants were grown in a greenhouse for 5–6 weeks under a 16 h day at 25 °C and 8 h night at 22 °C, and the two largest leaves of four-leaf-stage *N. benthamiana* plants were used for VIGS. Etiolated soybean seedlings were grown at 25 °C without light for 5–6 days before inoculation. *P. sojae* (P6497) and *P. capsici* (Pc35) strains were routinely maintained on 10 % vegetable (V8) juice medium at 25 °C in the dark.

***P. sojae* transformation and transcript level detection**. The sgRNAs design and transformation of *P. sojae* were carried out as the following protocol[47]. The sgRNA target sites were selected online (http://www.broadinstitute.org/rnai/public/analysis-tools/sgrna-design). The potential off-target sites were examined using the FungiDB (www.fungidb.org) alignment search tool (BLASTN) against the *P. sojae* genome and visual inspection of the results. *P. sojae* strain P6497 was maintained on V8 agar. Inoculate *P. sojae* discs onto nutrient pea agar plates for *P. sojae* transformation. Two-day-old mycelia, cultured in the nutrient pea broth liquid medium, were washed in 0.8 M mannitol, then placed in enzyme solution, and incubated for 40 min at room temperature with gentle shaking. The *P. sojae* protoplasts were collected by centrifugation at 1500 r.p.m. for 4 min and resuspended in W5 solution. 30 min later, the protoplasts were collected and resuspended in an equal volume of MMg solution to allow protoplasts to swell. For each of 1 mL protoplasts, 20–30 μg transforming DNA was added and incubated for 10 min on ice. Then, three successive aliquots of 580 mL each of freshly made polyethylene glycol (PEG) solution were pipetted into the protoplast suspension and gently mixed. After 20 min incubation on ice, 2, 8, 10 mL pea broth containing 0.5 M mannitol were added in turn, and the protoplasts were incubated overnight to regenerate in the dark. The regenerated protoplasts were suspended in liquid pea agar containing 0.5 M mannitol and 25 μg/mL G418 and plated. The visible colonies could be observed after 2–3 days incubation at 25 °C. The transformants were selected on the basis of green fluorescence, total genomic DNA (gDNA) was extracted from these fluorescent transformants. The *PsAvr3c* specific primers (Supplementary Table 1) were used to amplify the *PsAvr3c* gene from transformants gDNA and cloned into the T-simple plasmid for sequencing.

To determine the transcript levels of *PsAvr3c*, *PsXEG1*, and *PsojNIP* during *P. sojae* infection, total RNA was extracted from *P. sojae* P6497 zoospores and from infected susceptible soybean (williams) roots at 0.5, 1, 2, 3, 6, 24, 36 hpi. Transcript levels were measured by qRT-PCR using the *P. sojae* actin gene (VMD GeneID: 108986) as an internal reference.

**Plant inoculations**. The ability of *PsAvr3c* knockout mutants to infect soybean seedlings carrying *Rps3c* (L92-7857), *Rps3b* (PRX146-36) or *rps3c* (Hefeng47) were evaluated by hypocotyl inoculation with *P. sojae* mycelium, the wild type and mutant *P. sojae* strains were grown on V8 plates without G418 selection for 5 days. Infection photographs were taken at 48 h after inoculation, the lesion lengths were

measured at 48 hpi. Primers (Supplementary Table 1) specific for *P. sojae* actin gene (VMD GeneID: 108986) and soybean actin genes *CYP2* were used to quantify the relative biomass of *P. sojae* by qRT-PCR. PCR reactions were performed on an ABI Prism 7500 Fast real-time PCR System (Applied Biosystems, Foster City, CA, USA).

For assays of *P. capsici* infection on *N. benthamiana* leaves, overexpression of GFP-PsAvr3c, GFP-GmSKRP1/2, or GFP-PsAvr3c mutant proteins were confirmed by western blotting using anti-GFP antibody, and the pBinGFP2 empty vector was used as the negative control. The lesion areas (cm$^2$) were measured at 30 h under UV light after inoculation by mycelium. The primers (Supplementary Table 1) specific for *P. capsici* and *N. benthamiana* actin genes were used to quantify the relative biomass of pathogen by quantitative PCR.

**Yeast two-hybrid screening**. A Y2H screen with pGBKT7-PsAvr3c was performed as following described[48]. *PsAvr3c* gene without the signal peptide was cloned into the yeast vector pGBKT7 (Clontech). The soybean (*Glycine max*, cv. William 82) cDNA library was constructed in pGADT7 using total RNA extracted from soybean hypocotyl tissues collected 12 and 24 h after inoculated with *P. sojae* zoospores (Clontech). More than $6 \times 10^6$ primary yeast clones (three times screening will cover the library) were screened using pGBKT7-PsAvr3c as the baits. Potential yeast transformants containing cDNA clones interacting with PsAvr3c were selected using the SD/-Trp/-Leu/-His/-Ade selective medium.

**Confocal microscopy**. Patches of agro-infiltrated *N. benthamiana* leaves were cut and mounted in distilled water and analyzed using an LSM 710 laser scanning microscope (Carl Zeiss, Germany) with a ×20, ×40, or ×60 objective lens. The green and red fluorescence were observed at excitations of 488 nm or 561 nm, respectively.

**In vitro GST pull-down assays**. *PsAvr3c* gene without the regions encoding the signal peptide and RXLR-dEER motif was inserted into the pET32a vector (containing His tag), *GmSKRPs* and *NbSKRP* were inserted into the pGEX-4T-2 vector (containing GST tag) (GE Healthcare Life Science). pET32a empty vector, His-PsAvr3c, GST empty vector, GST-GmSKRPs, GST-NbSKRP were expressed in *E. coli* strain Rosseta2 respectively. The pull-down assay was performed using ProFound pull-down GST protein–protein interaction kit (Pierce) according to the manufacturer's instructions. The soluble total GST-fusion proteins were incubated with 25 μl glutathione agarose beads (Invitrogen) for 5 h at 4 °C. The beads were washed three times and then incubated with 1 mL bacterial lysates containing His proteins for another 3 h at 4 °C. Then beads were washed three times again, the presence of His proteins was detected by western blot using anti-His antibody.

**Co-immunoprecipitation assays**. The *PsAvr3c* gene without the signal peptide was inserted into pICH86988 using a one-step cloning kit (Vazyme Biotech), and *GmSKRPs* were inserted into pBinGFP2 for expression in *N. benthamiana*. Leaves of 6-week-old *N. benthamiana* plants were agro-infiltrated with FLAG-tagged PsAvr3c, GFP-tagged GmSKRPs (GFP as the control) and the P19 silencing suppressor. Two days after agro-infiltration, the leaves were frozen in liquid nitrogen and ground to powder using the mortar and pestle. Plant nuclei were isolation using Plant Nuclei Isolation/Extraction Kit (SIGMA, Product Code CELLYTPN1), nuclear protein were extracted using lysis buffer [1 mM ethylene-diaminetetraacetic acid (EDTA), 1% Triton,10 mM Tris (pH 8.0), 150 mM NaCl] plus 1 mM phenylmethylsulfonyl fluoride (PMSF) and a protease inhibitor cocktail (Sigma-Aldrich). The samples were centrifuged at 4 °C for 10 min at $12,000 \times g$ and the supernatant was transferred to a new tube. For GFP-IP, 1 mL of supernatant was incubated at 4 °C for 3–4 h with 25 μL of GFP-Trap_A beads (Chromotek, Planegg-Martinsried, Germany). The beads were then collected by centrifugation at $2500 \times g$ and washed four times in 1 mL of washing buffer [10 mM Tris–Cl (pH 7.5), 150 mM NaCl and 0.5 mM EDTA]. Bound proteins were boiled for 10 min at 95 °C. The beads can be collected by centrifugation at 2500 x g for 2 min at 4 °C and SDS-PAGE is performed with the supernatant, and the presence of FLAG proteins was detected by western blot using anti-FLAG M2-Peroxidase (HRP) antibody (1:2000; #A8592; Sigma-Aldrich). The images were caught by Tanon-5200 Muti Chemiluminescent Imaging System (Tanon, China).

**Western blotting**. The separated proteins were transferred from the gel to PVDF membrane and then blocked using PBST (PBS with 0.1% Tween 20) containing 5% non-fat milk for 30 min at room temperature with 60 r.p.m. shaking, anti-GST (1:5000; #M20007; Abmart), anti-his (1:5000; #M30111; Abmart), Anti-Flag (1:5000; #F3165; Sigma-Aldrich), anti-GFP (1:5000; #M20004; Abmart), anti-RFP (1:5000; #5f8; Chromotek), antibodies were added to PBSTM (PBS with 0.1% Tween 20 and 5% non-fat dry milk) and incubated at room temperature for 3–4 h, followed by three times washes with PBST. The membrane was then incubated with a goat anti-mouse IRDye 800CW antibody (Odyssey, no. 926-32210; Li-Cor) at a ratio of 1:10,000 in PBSTM at room temperature for 30 min with 60 r.p.m. shaking. The membrane was washed three times with PBST, and then visualized by excitation at 800 nm. Full-size images are presented in Supplementary Fig. 21.

**VIGS of *NbSKRP* gene in *N. benthamiana***. We used the Tobacco Rattle Virus (TRV)-based VIGS system[49] to silence *NbSKRP* genes (Supplementary Table 3) in *N. benthamiana*. Fragments from each end of the *NbSKRP* gene were cloned into the pTRV2 vector to generate silencing constructs TRV:NbSKRP-5′ and TRV:NbSKRP-3′. The TRV:GFP vector was used as a control in this assay. Primer sequences are shown in Supplementary Table 1. The four-leaf stage *N. benthamiana* plants were infiltrated with GV3101 *A. tumefaciens* strains containing a mixture of pTRV1 and each NbSKRP VIGS construct or the GFP control at $OD_{600}$ = 0.5 each. The fully expanded six leaves of the silenced plants were then used for inoculation and quantitative PCR assays.

**Soybean transformation and *P. sojae* infection assays**. The soybean transformation and *P. sojae* infection assays were carried out as described[25]. For overexpression assay, soybean cotyledons were inoculated with *A. rhizogenes* strain K599 carrying GFP, GFP-GmSKRP1/2, GFP-PsAvr3c. The overexpression hairy roots were selected on the basis of green fluorescence and infected by *P. sojae* strain P6497 expressing RFP (RFP-P6497), expression of GFP-GmSKRP1/2, GFP-PsAvr3c or GFP proteins in hairy roots were confirmed by western blotting using anti-GFP antibody. Relative numbers of oospores produced on infected hairy roots were measured at 48 hpi and normalized to GFP. For biomass quantification, total DNA was extracted from inoculated hairy roots at 48 hpi and primers specific for soybean housekeeping gene CYP2 and *P. sojae* actin gene were used for qRT-PCR.

For the gene-silencing assay, we cloned a 162 bp fragment at the 5′ end and a 122 bp fragment at the 3′ end of GmSKRP1, respectively. These two fragments were individually cloned into the vector PK7GWIWG2D(II) to generate hairpin silencing constructs, GmSKRPs-5′ and GmSKRPs-3′. The silenced hairy roots were selected based on red fluorescence. Soybean cotyledons were inoculated with *A. rhizogenes* strain K599 carrying PK7GWIWG2D(II) empty vector, PK7GWIWG2D(II)-GmSKRPs-5′ and PK7GWIWG2D(II)-GmSKRPs-3′. Primer sequences are shown in Supplementary Table 1. The silencing hairy roots were selected on the basis of red fluorescence and infected by *P. sojae* strain P6497 expressing GFP (GFP-P6497), silencing efficiency of GmSKRP1/2 in hairy roots was confirmed by quantitative PCR. Relative numbers of oospores produced on infected hairy roots were measured at 48 hpi and normalized to empty vector.

**RNA-seq data analysis and AS event identification**. Transcription data for *PsAvr3c* as obtained from previous RNA-seq data[25]. Alternative splicing data were obtained from RNA-seq of overexpression GFP, GFP-GmSKRP1 and GFP-PsAvr3c soybean hairy roots (cultivar: Williams 82). The total RNA was extracted using TRIZOL reagent (Invitrogen Cat#15596026) and used to make RNA-seq libraries with the Illumina TruSeq RNA Sample Preparation Kit v2 following the manufacturer's recommendations. These libraries were sequenced using Illumina HiSeq 4000 in paired-end mode with a read length of 150 bp (1gene, Hang Zhou, China).

The raw RNA-seq reads generated from Illumina sequence platform were filtered using the ht2-filter from HTQC package (1.92.1) with the default parameters to remove low-quality reads[50]. Then filtered reads were aligned to the soybean genome (Gmax_275_v2.0) using tophat-2.0.11 with anchor length >8 nt for spliced alignments. Above mapped reads were used and assembled with Cufflinks v2.1.1[51] using following option: -F 0.05 -A 0.01 -I 100,000 min-intron-length 30. The output GTF files from Cufflinks and the BAM files from tophat were loaded into rMATS.3.2.2[52] to identify the differential alternative splicing events using following parameters: -t paired -len 125 -a 8 -c 0.0001 -analysis U. In total, alternative splicing events with $P$ value < 0.05 were regarded as the differential alternative splicing events.

**RT-PCR analyses and measurement of splicing variant ratio**. RT-PCR analyses were performed with cDNA prepared from different overexpression soybean hairy roots and different infection soybean roots with Takara EX Taq$^{TM}$ polymerase and gene specific primers (Supplementary Table 1). An equal amount of cDNA template in each reaction was verified by amplifying soybean actin gene CYP2, and then PCR reaction with primers specific to each gene was conducted. The intensities of PCR products were quantified by ImageJ software, and the relative abundance were then calculated by using bottom (or top) bands in the left electrophoresis line as reference (set to 1). Splicing ratio measurements were conducted as previously reported[53]. Splicing ratio was calculated by determining the level of spliced RNA normalized to the level of unspliced RNA. The spliced primers (Supplementary Table 1) were designed crossed the exon-exon junction and the unspliced primers (Supplementary Table 1) were designed to span the intron–exon junction. The data were shown as average fold-change over the splicing ratio from three biological repeats.

**Motif prediction**. The signal peptide of PsAvr3c was predicted according to the website (http://www.cbs.dtu.dk/services/SignalP/). The proteins NLS motifs and NAC, WRKY domains in the paper are predicted according to Prosite (http://prosite.expasy.org/). Two W-motifs of PsAvr3c were reported earlier by Dong and his colleagues[23]. NCBI conserved domain database (CDD) (https://www.ncbi.nlm.nih.gov/cdd/) was used to identify any domain or motif based on GmSKRP1/2 proteins sequence.

**Isothermal titration calorimetry (ITC)**. Calorimetry experiments were carried out at 25 °C in 20 mM HEPES pH 7.5, 200 mM NaCl, using a PEAQ-ITC instrument. For protein–protein interactions, the calorimetric cell was filled with 40 μM GmSKRP1 and titrated with 400 μM PsAvr3c from the syringe. A single injection of 0.4 μl of PsAvr3c was followed by 19 injections of 2 μl each. Injections were made at 150 s intervals with a stirring speed of 750 r.p.m. The raw titration data were integrated and fitted to a one-site-binding model using the MicroCal PEAQ-ITC analysis software.

**Data availability**. The RNA-seq data that support the finding of this study have been deposited in Gene Expression Omnibus (GEO) database (https://www.ncbi.nlm.nih.gov/geo/) with the accession codes GSE100985. The authors declare that the other data supporting the findings of this study are available from the corresponding author upon request.

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

## Acknowledgements

We thank Dr. Meixiang Zhang (Nanjing Agricultural University) for providing *P. capsici* strains PC35, Dr. Danyu Shen for providing sequence of PsAvr3c-like proteins, Dr. Edgar Huitema (James Hutton Institute) for protein extraction assistance, Prof. Xiaorong Tao (Nanjing Agricultural University) and Prof. Brett Tyler (Oregon State University) for helpful discussions, Mr. Yachun Lin and Ms. Ling Chen (Nanjing Agricultural University) for microscopy work, Mr. Yao Zhao and Ms. Yu Hang (Nanjing Agricultural University) for soybean transformation work. This work is supported by the Chinese National Science Fund for Excellent Young Scholars (31422044), Chinese Thousand Talents Plan, Agriculture and Agri-Food Canada Genomics Research and Development Initiative, and the Fundamental Research Funds for the Central Universities (KYTZ201403).

## Author contributions

J.H., Y.Z., B.D.G., M.Q. performed experiments; G.H.K., L.K., L.F.G., W.W.Y., Z.W., Y. W., M.F.J., W.M.X. analyzed data; J.H., Y.Z., T.X.Y. contributed constructs; J.H., L.F.G., Z.W., Z.G.Z., X.B.Z., Y.C.W., S.D. designed experiments; J.H., M.G., S.D. wrote the manuscript; all authors commented on the article before submission.

## Additional information

**Competing interests:** The authors declare no competing financial interests.

