## [Peer Review File · Nature Communications]

Reviewers' comments:

Reviewer #1 (Remarks to the Author):

In this work, Huang and authors, identified two (related) host targets for the *Phytophthora sojae* effector AVR3c in soybean. The authors first show that deletion of Avr3c in *P. sojae* leads to evasion of ETI, conferred by Rps3c and negatively impacts virulence on soybean plants that lack Rps3c. These results, combined with the observation that Avr3c is present and expressed in all strains examined to date, leads the authors to suggest that this RXLR effector is essential for pathogen virulence.

The authors then use Y2H to identify two proteins from soybean (GmSRKP1 and GmSRKP2) and confirm this interaction by in vitro and in vivo pull-downs. Subsequent functional analyses (silencing and over expression in *N. benthamiana* leaves and soybean hairy roots) is then used to demonstrate that these proteins help specify the levels of immunity or susceptibility to *P. capsici* and *P. sojae* respectively. Over expression of GmSRKP1/2 promotes virulence whereas silencing appears to tip the scale in favour of the host.

Finally and to elucidate the roles of GmSRKP1/2 and AVR3c in immunity and virulence, the authors embarked on functional analyses of the AVR3c host targets. GmSRKP1/2 interact with spliceosome factors and this interaction appears to be critical for virulence. As GmSRKP1/2 are considered splicing factors, the authors over-expressed GmSRKP1 & GmSRKP2 as well as AVR3c and assessed whether these proteins impacted on pre-mRNA (alternative) splicing. These analyses yield a large number of candidate target genes (>1,000), some of which are thought to have roles in immunity. From these analyses, the authors conclude that AVR3c targets alternative splicing in plants to subvert immunity.

Overall, this work represents a major advance in the field of plant-microbe interactions. Although many examples exist of effector-mediated reprogramming of host processes, to my knowledge this is the first report that shows a pathogen targeting splicing in plants. Given the widely accepted view that alternative splicing underpins the regulation of important cellular processes in eukaryotes (including plant immunity), the findings presented by the authors will greatly impact the field of plant-microbe interactions, (alternative) mRNA splicing and immunity.

Importantly, the MS as presented, lacks in certain areas, which in my view requires attention. My concerns, comments and suggestions are listed below.

1. Knockout of Avr3c and virulence assays. The authors show that by using CRISPR/Cas9, Avr3c is knocked out and homozygous lines are created. According to Dong et al (PlosOne 2009) Avr3c is a multi copy gene with 2 identical copies present in close proximity. The authors should mention this in the text and demonstrate that both copies are knocked out (or not expressed anymore). Furthermore, the virulence data (loss of avirulence, fig 1b) are not convincing, as the phenotypes are not clear in such small panels. Bigger panels and quantification of growth would be desirable if the authors wish to make the point that Rps3c recognises Avr3c. For the purpose of this study, I am not sure whether the

avirulence data is needed (unless a connection is made between Rps3c function and alternative splicing, see below). Rather, a complementation assay (restoration of *P. sojae* mutants' virulence on roots expressing Avr3c) is more useful in the context of the main message as it would help demonstrate that Avr3c is the only virulence factor that has been disrupted. Figure 1e & 5b needs modification. Oospores need to be identifiable by the reader (arrowheads and bigger panel?).

2. The authors identify GmSRKP1/2 as candidate host targets for Avr3c. Over-expression promotes virulence whereas silencing enhances immunity. Based on these observations, the authors state that GmSRKP1/2 are susceptibility factors. I disagree with this statement. To me GmSRKP1/2 appear to act as negative regulators of immunity. Naming them as such is more appropriate, given the subsequent claim that by targeting these proteins, immunity is subverted. In my view, susceptibility factors are proteins that aid pathogen infection in a way that is independent from immunity (e.g. increase nutrient availability).

3. The authors state that GmSRKP1-spliceosome association is required for susceptibility. By saying this, the authors imply that this is connected to the activity of AVR3c. However, for this statement to stand up to scrutiny, the authors should demonstrate that AVR3c can still interact with GmSRKP1. If it still does, the authors statement is likely to be true, if not, the alternative hypothesis is that Avr3c-GmSRKP1 interaction is required for virulence. Alternatively, the statement should be modified such that it conveys the message that regulation of immunity requires this association.

4. The authors conclude their work by demonstrating that GmSRKP1 has a role in alternative splicing by identifying genes that are more or less alternatively spliced in the presence of GmSRKP1. The authors then go on to show that during infection and upon over expression of Avr3c, these alternative splicing events do take place. In addition, AS is firmly connected to AVR3c function as Avr3c mutants do not appear able to trigger these AS events during infection. Given the title of this MS and the claims made, I feel that these are the standout result that warrants expansion, which in turn, would justify the claim made. For example, a more comprehensive study using the AS data could be embarked on by: (i) Assessing whether there is significant enrichment of defense-associated genes amongst AS targets or perhaps in those AS events that result in inactivation of a given protein, (ii) demonstrate that GmSRKP1 and GmSRKP2 indeed interact with the target genes of interest or alternatively, assess whether the presence of Avr3c changes GmSRKP1/2 specificity towards their mRNA targets (which one may expect based on the observation that co-expression of Avr3c and GmSRKP1/2 gives an additional boost to infection, (iii) functionally validate the role of NAC-TF and/or WRKY transcription factors in immunity to *P. sojae*. The latter is a high-risk experiment however, given the number of possible candidate targets that could underpin the phenotype. AS of >1,000 could mean that the enhanced susceptibility phenotype is due to multiple target genes or only a few major players. In either scenario, the chances of identifying a robust phenotype that connects GmSRKP1/2 function to a target, its fate and susceptibility is small. It is for that reason that the selection of two examples from >1,000 candidates is subjective and too suggestive.

5. The authors present numerous pieces of data for which I either question relevance, do not see a connection to the model or is open for alternative interpretation. For example, the authors claim that

Avr3c over-expression re-locates GmSRKP1/2 to the nucleolus. Firstly, re-location suggests that protein present in the nucleoplasm is moved to the nucleolus, which is not correct as the nucleolar pool could be newly synthesized protein, Avr3c could prevent GmSRKP1/2 export from the nucleolus to the nucleoplasm and so on. Secondly, a connection is not evident between a change in localisation and GmSRKP1/2 altered function. The model (fig 7) does not mention this. The same goes for the prevention of nuclear localisation with Nuclear export signals, mutated NLS sequences etc. it is an extensive analyses that does not add much to the major finding. The authors use *P. capsici* in a number of experiments to validate or confirm results. Although of great value (see point below), again this distracts the reader from the main message. In short, I thus recommend for the authors to simplify this work by cutting out the least relevant pieces of data and strengthen the major conclusion by adding work supporting the major claim.

6. The discussion could be more focussed around the model and pose additional questions. For example, Avr3c mediated changes in AS are discussed, but the exact mechanisms are not explored. Avr3c could act as a factor that helps recruit or bind SFs to pre-mRNAs or alternatively, change GmSRKP1/2 specificity. I find the “prevention of degradation” option a possibility, but not the most likely scenario. Rather, I suspect that lower turnover is due to retention of GmSRKP1/2 to pre-mRNA-spliceosome complexes. The authors use Rps3c lines to demonstrate that Avr3c is knocked out and leads to evasion of ETI. If true, either GmSRKP1/2 are guarded of Rps3c or alternatively (and perhaps more likely) an alternative splicing event on one or more mRNAs leads to a signal that triggers Rps3c mediated immunity. To my surprise, the connection to Rps3c is not discussed at all. I believe this to be of interest as the activation of AS by *P. capsici* and the presence of Avr3c homologs in other Phytophthora species, raises the prospect that Rps3c could be (modified and) deployed successfully against other Phytophthoras. This should either be mentioned in the discussion or the authors may as well remove all the work with the other Phytophthora species mentioned (homologs, *P. capsici* etc).

7. The text contains a fair number of errors. A detailed edit and check is required. For example, Lines 227-230 refer to the wrong supplementary figure and there are grammatical errors that need correction.

8. Supplementary figure 13a. There is a thin line running through the western image. If it is meant to indicate an image modification, they should be more explicitly separated (together with the CBB stained panel below it).

Reviewer #2 (Remarks to the Author):

Comments for the Authors

In this interesting manuscript, Huang et al expand our knowledge of a largely unexplored field: host-pathogen interactions at the level of alternative splicing. The authors start with the characterization of Phytophthora avirulence effector PsAvr3c and investigate its role in the pathogenesis in soybean. They show that PsAvr3c is a nuclear protein that binds to two soybean paralogous uncharacterized proteins,

named GmSRKP1 and GmSRKP2, thus leading to their stabilization. The authors address functions of these proteins and demonstrate that they are important for plant immunity. Remarkably, the authors show that these proteins interact with several RNA binding proteins including orthologs of well-characterized Arabidopsis splicing factors SR45, U2AF35 and U1-70K, suggesting a role for GmSRKP1 and GmSRKP2 in pre-mRNA splicing. To get further insights into GmSRKP1/2 functions, the authors analyzed splicing changes using RNA-seq of GFP-GmSRKP1 over-expression lines and a GFP expressing control. The analysis identifies changes in several basic types of alternative splicing events in multiple genes, including NAC and WRKY transcription factors potentially involved in pathogen response. Moreover, the authors show that overexpression of PsAvr3c effector or its mutant version corroborates changes in splicing efficiencies of NAC and WRKY genes.

Though the authors present many solid results that are interesting for the plant community, there are several issues I found problematic as follows:

I may have missed it, but RNA-seq analysis was performed on two independent GFP-GmSRKP1 over-expression lines, which represented two biological replicates, and one GFP control line (in one replicate). The use of replicates (also for control) is essential for detecting differential alternative splicing. There is a high variability among two GFP-GmSRKP1 replicates as reflected by low number of the overlapping AS events. As the level of variability among replicates is so high, more replicates both for GFP-GmSRKP1 and importantly for GFP control are needed to deliver reliable lists of affected genes and AS events.

The authors constructed a soybean transcript dataset with Cufflinks that was used as an input for rMATS to quantify differential alternative splicing. Currently, accurate reconstruction of transcript isoforms from Illumina RNA-seq short reads remains a significant challenge. For example, Steijger et al (Assessment of transcript reconstruction methods for RNA-seq. *Nature Methods* 2013;10(12) 1177-84) demonstrate that Cufflinks fails to recognize 40-60% of exons leading to mis-assembled and missing transcript isoforms thus resulting in poor quantification of alternative splicing. Therefore, thorough experimental validation is required to evaluate and support RNA-seq analysis. The authors performed RT-qPCRs for two putative pathogen defense genes, NAC and WRKY, to validate AS changes detected by RNA-seq analysis. Firstly, this is a biased approach. Secondly, two genes are not sufficient to evaluate the validation rate. Validation set should be expanded and should include random genes/AS events with a broad range of AS changes.

Throughout the paper, the authors draw a link to Ser/Arg-rich (SR) proteins, based on the fact that GmSRKP1 and GmSRKP2 are rich in serines, lysines and arginines and that they interact with SR-like protein SR45. Bona fide SR proteins have characteristic features such as one or two RRM at the N-terminus followed by a region rich in RS dipeptides. The precise definition and the unified nomenclature for plant and animal SR proteins have been established by the splicing community to avoid confusion and are provided for plants in Barta A, Kalyna M, Reddy AS. Implementing a rational and consistent nomenclature for serine/arginine-rich protein splicing factors (SR proteins) in plants. *Plant Cell*. 2010 Sep;22(9):2926-9 and for human in Manley J.L., Krainer A.R. (2010). A rational nomenclature for serine/arginine-rich protein splicing factors (SR proteins). *Genes Dev*. 24: 1073–1074. According to the established consensus SR45 is an SR-like protein and does not belong to SR proteins. In addition,

GmRBPs and NbrBP (also described as SR proteins – e.g. Supplementary Figure 16) do not belong SR family and represent SR-like proteins (potentially, homologs of SR45a/tra-2 as I can judge from a quick BLAST search). Moreover, GmSRKP1 and GmSRKP2 cannot be considered as “unconventional SR proteins” (line 125) as they do not have any features of SR or SR-like proteins. Therefore, all statements regarding SR proteins have to be revised and removed. For example, introduction (78-82, 125), results (348-351, 358, 376-378), discussion (543-546), Supplementary Figures 16 and 17b. List of references also needs a revision in this regard. To avoid any further confusion in the field, I would suggest renaming GmSRKP1 and GmSRKP2 to GmSKRP1 and GmSKRP1 as they are richer in lysines than arginines.

The description of RNA-seq results and of the effect of GmSKRP1 on differential splicing is ambiguous, or at least unclear. Terms “genes”, “transcripts”, “mRNA transcripts”, “pre-mRNA transcripts” are used interchangeably in abstract and in the text (e.g. Abstract 34-35, Introduction 128 “more than one thousand mRNA transcripts”; Results 402 “>1,000 pre-mRNA transcripts”; Discussion 497 “>1,000 soybean genes” and 506 “Among the >1,000 genes identified”). Based on data in the Supplementary Table 2 I think that the authors have data on >1,000 AS events. The table contains 1108 entries, each corresponding to an AS event. There are 922 genes in this table, among them 142 genes have more than one AS event. From the data presented, it is not clear how many AS transcripts are generated, because an AS transcript can contain more than one AS event.

Other points/suggestions:

64-65 “RNA splicing edits pre-mRNA transcripts by removing introns from nascent mRNA transcripts and joining exons together” and other similar sentences in the text - mRNA (messenger RNA) is a spliced transcript, introns are removed from pre-mRNA.

66 “This process (splicing) takes place within the nucleus either co-transcriptionally or immediately after transcription.” – considerable delay in splicing is widespread, e.g. in intron retention transcripts (Boothby 2013, Wong 2013, Shalgi 2014).

67-69 “Alternative splicing (AS) generates multiple transcripts from a single gene and can result from intron retention, exon skipping, or selection of an alternative 5’ donor sites or 3’ acceptor site” – AS is not limited to these events. In addition to mutually exclusive exons, which also represent basic AS events, there are non-canonical AS events such as microexons and exitrons (Irimia et al. Cell. 2014 Dec 18;159(7):1511-23 and Marquez et al. Genome Res. 2015 Jul;25(7):995-1007)

69-70 references 7 and 8 - replace with the newer, less specific reference, for example Reddy et al. Complexity of the alternative splicing landscape in plants. Plant Cell. 2013 Oct;25(10):3657-83. doi: 10.1105/tpc.113.117523

72 wrong reference for Arabidopsis – replace with Marquez et al. Genome Res. 2012 Jun;22(6):1184-95

93-94 “Another example is provided by the RCT1 gene of Colletotrichum trifolii, which confers resistance against multiple races of fungal pathogen” - it is a Medicago truncatula gene, which confers resistance to Colletotrichum

102 reference 15 – replace with the reference to the original paper Yang et al Proc Natl Acad Sci U S A. 2008 Aug 26; 105(34): 12164–12169.

139 “The qRT-PCR data demonstrated that the expression of PsAvr3c is induced at early stages of infection and is greatest at 1 hpi, which is consistent with RNA-seq data (Supplementary Fig. 1a, 1b)” - I may have missed it, but I did not find this RNA-seq analysis (also no description in Methods)

217 Sequences of GmSRKP1/2 (and NbSRKP) should be added to Supplementary Table 3, which contains sequences of all proteins mentioned in the manuscript except these.

219 “GmSRKP1 and GmSRKP2 are 558 bp and 552 bp in length, respectively” – these are the lengths of the coding regions of these genes. The genes themselves are longer as they contain introns and UTRs.

258 “We observed that RFP-GmSRKP1/2 fusion proteins predominantly accumulated in the nucleoplasm, with a low proportion of nuclear speckle localization.” – I do not see any speckles.

432 “alternative splicing of the two predicted transcriptional factor genes in the presence of PsAvr3c effector results in emergence of premature stop codons in retained intron sequences. Thus, the transcripts are likely be degraded by non-sense mediated RNA decay or translated as truncated nonfunctional proteins” and 524-526 – recent evidence implicates that intron retention transcripts are not degraded by nonsense-mediated mRNA decay (Kalyna et al. Nucleic Acids Res. 2012 Mar;40(6):2454-69, Leviatan et al. PLoS ONE 8, e66511 (2013) and papers from human/mammalian field) as they are retained in the nucleus and therefore are not accessible to NMD machinery (Gohring et al. Plant Cell. 2014 Feb;26(2):754-64 and other papers, also from human/mammalian field – Boothby et al 2013, Wong et al 2013, Shalgi et al 2014).

455 “Genes encoding SRKP-like proteins are prevalent in plants” – 1) no data is presented to support this statement; 2) are SRKPs plant-specific?

459 “pathogen effector interferes with host RNA splicing machinery to reprogram the processing mRNA transcripts, and we propose that pathogens suppress host immunity at mRNA modification layer.” – 1) to reprogram the splicing of pre-mRNA (mRNA transcripts are spliced products); 2) RNA modification – this term is mostly applied for changes to the chemical composition of RNA (capping, m6A, m5C, pseudouridine etc), aka “epitranscriptome”.

467 “A KH type RBP protein” – it says the same thing twice: RBP protein = RNA binding protein protein

472 reference 33 is not complete

513 “...mRNA sequencing is not sensitive enough to detect the changes in splicing for many genes, as other authors have speculated previously³⁴. Therefore, the actual number of affected splicing events could be greater than we identified in this initial analysis.” – the problem is not in the sensitivity of RNA-seq, which with sufficient sequencing depth can detect even minor unspliced pre-mRNAs, but in RNA-seq analyses based on assemblies of transcripts from short reads, miss-classification of complex AS events and on low sensitivity and precision of quantification programs.

542 and throughout the text – I would suggest to avoid using superscript in U2AF35 and to replace it with U2AF35, in order to avoid confusion with the references.

Methods:

- Plant growth conditions are not described properly (different species, different experiments)
- No RNA-seq description for data presented in Supplementary Fig. 1.
- RNA-seq paragraph is lacking essential descriptions, e.g. no description of RNA isolation, library preparation, RNA-seq read characteristics (length, single or paired end?), no information on soybean genome used for read mapping.

Figures:

Figure 7 is not mentioned in the text.

1147 Fig. 7 legend “In particular, some defense related genes generate alternative splicing isoform and yield truncated nonfunctional proteins” – there is no evidence that they yield any proteins. In addition, truncated proteins can be functional.

Reviewer #3 (Remarks to the Author):

The manuscript “An oomycete plant pathogen reprograms host pre-mRNA splicing to subvert immunity” by Huang et al. describes a functional characterization of the effector PsAvr3c from *Phytophthora sojae*, a soybean pathogen. In a large number of well-controlled and high-quality experiments, the authors show that PsAvr3c binds to GmSRKP1/2, which are serine/arginine/lysine rich proteins of unknown functions. The authors further demonstrate that this interaction results in GmSRKP stabilization and partial relocalization into the nucleolus. Using immune precipitation and mass spectrometry, interacting partners of SRKPs are identified, including several splicing factors. Finally, splicing patterns upon SRKP overexpression are analyzed in a transcriptome-wide manner. Two of the events within defense genes are independently verified; furthermore, some changes in these events are also found upon infection with *P. sojae* in an PsAvr3c effector-dependent manner.

This work provides a model of effector-mediated changes in alternative splicing as critical component of virulence function. This would be indeed a very interesting mechanism. The manuscript includes many interesting data, with the only major weakness in the analysis of the alternative splicing pattern changes. First, it would have been more interesting to analyze transcriptome-wide AS in plants infected with *P. sojae* with functional and non-functional Avr3c. Second, the number of validated events is very low, and any correlation between SRKP overexpression and the infected samples is not convincing. For details see below. I think the authors should extend some of these analyses, and clearly highlight the open questions. With these edits, I would see this work as an important contribution to the field.

Major comments:

1) The splicing analyses in Fig. 6 show changes in the splicing ratios. These changes can result from alternative splicing or altered stability of one of the isoforms. Previous work had shown inactivation of NMD in plants upon pathogen infection. Accordingly, the changes observed in this study might result from NMD inactivation. In line with this, intron-retained transcripts with NMD target features over-accumulate upon GFP-SRKP1 expression. To provide evidence for changes in splicing, the authors need to separately quantify both splicing forms (intron-spliced and intron-retained); if changes occur on the level of splicing, opposite changes in the two isoforms are expected.

2) I was wondering why the authors haven't performed RNA-Seq from soybean plants infected with the different *P. sojae* strains (analyzed in Fig. 6e). This is expected to result in more physiological data than the analysis of the overexpression lines, also given that redistribution of SRKP1 in an effector-dependent manner was observed. Furthermore, Fig. 6e lacks a control with a strain carrying an unrelated effector, such as PsAvh52. Relative to this control, the WT and T71 are expected to result in a decrease of the splicing efficiency, while the knockouts should show splicing patterns similar to the control. In the absence of RNA-Seq data from these samples, it would be more convincing to analyze instead of only two some more candidates (at least 5 more randomly chosen candidates from the event list of the overexpression lines).

3) I find the observation of SRKP1/2 relocation to the nucleolus in the presence of Avr3c interesting in light of a splicing regulatory effect. As a possible explanation, SRKP1/2-mediated splicing function might be suppressed in the presence of Avr3c by removing these factors from the sites of splicing (nucleoplasm and speckles). The authors should discuss this possibility and clearly state that splicing is not expected to occur in the nucleolus.

Further comments:

1) The soybean infection assay shown in Fig. 1b should be explained in more detail, in particular the interpretation of the results. The replicates should be included in Supplement. Could the spreading zone also be quantified as in Fig. 1c?

2) Fig. 2d shows the results from isothermal titration calorimetry for measuring the binding affinity between PsAvr3c and GmSRKP1. The authors provide a K_d as a representative value of three independent experiments. It would be more convincing to show all three experiments and provide all three values or a mean K_d with standard deviation.

3) Some of the quantitative data in Supplement need better description, always provide number of replicates, types of error bars, and displayed values (mean?).

Minor points:

1) l. 249-251: "To determine whether greater accumulation of GmSRKP1/2 might be due to enhanced gene expression, qPCR was used to quantify GmSRKP1/2 transcriptional level." Statement should be changed to "transcript level" as steady state levels are measured, which can be influenced by the transcriptional rate and RNA turnover rate.

2) Fig. 7 is not mentioned in the text. Furthermore, some of the symbols like the ovals need to be defined. There is no evidence for translation of the alternative splicing products; the corresponding statement in the legend should be toned down or better removed.

Editor

Comments: we feel that a direct comparison of the influence of functional and non-functional Avr3c on the transcriptome (as suggested by reviewer #3) would greatly strengthen the case for further consideration.

Response: We really appreciate that you offered us an opportunity to revise our manuscript. As you point out, reviewers are positive about the novelty of our discovery but also raise some concerns. We read carefully through all the comments. They are fair and attention to many could improve the manuscript. Therefore, we extensively modified our manuscript based on these suggestions. More importantly, we added more experimental data to support our conclusions. These efforts included sequencing more biological samples, new genome-wide splicing analysis (Fig. 6), validating more splicing events by RT-PCR experiments (Fig. 7), and adding other experimental data as asked by reviewers. We hope our current version will meet all reviewers' concerns.

One of the suggestions from reviewer 3, which you also mention, is a direct comparison of the influence of functional and non-functional PsAvr3c on the soybean transcriptome. This is indeed a good suggestion. We agree that a direct comparison of RNA-seq data from infected soybeans (challenged with *P. sojae* wild type and PsAvr3c KO mutants) is a biologically appealing experiment. However, with due respect to the reviewer and editor, we would like to raise our own concerns with such an experiment and suggest an alternative.

Based on previous analyses, it is known that there are massive transcriptional changes during infection of soybean by *P. sojae*, which will create a lot of noise. More importantly, *Phytophthora sojae* secretes several hundred cytoplasmic effectors besides PsAvr3c (Tyler et al. Science, 2006, Jiang et al., PNAS, 2009), some of which may also directly or indirectly affect splicing. So, although the suggested experiment might work, we think that it is quite risky because the signal (PsAvr3c effects) could be lost in the noise (infection related changes that are independent of PsAvr3c). In fact, we do not know of any study to date that has successfully used such an approach to characterize the functionality of an oomycete virulence effector. (two relative publications for reviewers and editor to consider are: Jing M et al. Nat Commun. 2016; Boevink PC et al. Nat Commun. 2016). We also note that reviewer 1 and reviewer 2 do not suggest such an experiment, despite that they also would like to see the RNA-seq analysis improved.

To strengthen our manuscript, we suggest that ectopic expression of PsAvr3c in hairy roots is a preferable experiment than the infection site analysis. Our feeling is that it will provide a cleaner analysis of the effect of PsAvr3c on host cellular operations, since it is not confounded by the massive effects of infection and host defense responses. Ectopic expression in the plant is a commonly used approach to probe effector functionality in other host-pathogen systems, and to isolate effector-specific effects from other infection related processes. We performed a deep RNA-seq analysis with treatments such as expression of GmSKRP1, PsAvr3c, and GFP, and provide three biological replicates for each. Then we verified those interesting AS events by RT-PCR experiments, on a larger scale than we previously did, as reviewers suggested. We test 10 genes transcripts, identified as AS from the RNA-seq work, in overexpressing soybean lines (GFP, GmSKRP1, GmSKRP1^{mut}, PsAvr3c, PsAvr3c^{M4}) and *P. sojae* infected soybean tissues (three field strains and PsAvr3c KO mutants).

The major discovery from our work is a new model for how pathogens manipulate the host AS process. We hope that a deeper RNA-seq analysis based on more replicates together with an expanded set of AS validations will meet the reviewers' concerns and would be sufficient to support this model.

Reviewer #1 (Remarks to the Author):

In this work, Huang and authors, identified two (related) host targets for the *Phytophthora sojae* effector AVR3C in soybean. The authors first show that deletion of Avr3c in *P. sojae* leads to evasion of ETI, conferred by Rps3c and negatively impacts virulence on soybean plants that lack Rps3c. These results, combined with the observation that Avr3c is present and expressed in all strains examined to date, leads the authors to suggest that this RXLR effector is essential for pathogen virulence.

The authors then use Y2H to identify two proteins from soybean (GmSKRP1 and GmSKRP2) and confirm this interaction by in vitro and in vivo pull-downs. Subsequent functional analyses (silencing and over expression in *N. benthamiana* leaves and soybean hairy roots) is then used to demonstrate that these proteins help specify the levels of immunity or susceptibility to *P. capsici* and *P. sojae* respectively. Over expression of GmSKRP1/2 promotes virulence whereas silencing appears to tip the scale in favour of the host.

Finally and to elucidate the roles of GmSKRP1/2 and AVR3c in immunity and virulence, the authors embarked on functional analyses of the AVR3c host targets. GmSKRP1/2 interact with spliceosome factors and this interaction appears to be critical for virulence. As GmSKRP1/2 are considered splicing factors, the authors over-expressed GmSKRP1 & GmSKRP2 as well as AVR3c and assessed whether these proteins impacted on pre-mRNA (alternative) splicing. These analyses yield a large number of candidate target genes (>1,000), some of which are thought to have roles in immunity. From these analyses, the authors conclude that AVR3c targets alternative splicing in plants to subvert immunity.

Overall, this work represents a major advance in the field of plant-microbe interactions. Although many examples exist of effector-mediated reprogramming of host processes, to my knowledge this is the first report that shows a pathogen targeting splicing in plants. Given the widely accepted view that alternative splicing underpins the regulation of important cellular processes in eukaryotes (including plant immunity), the findings presented by the authors will greatly impact the field of plant-microbe interactions, (alternative) mRNA splicing and immunity.

Importantly, the MS as presented, lacks in certain areas, which in my view requires attention. My concerns, comments and suggestions are listed below.

Comments: Knockout of Avr3c and virulence assays. The authors show that by using CRISPR/Cas9, Avr3c is knocked out and homozygous lines are created. According to Dong et al (PlosOne 2009) Avr3c is a multi copy gene with 2 identical copies present in close proximity. The authors should mention this in the text and demonstrate that both copies are knocked out (or not expressed anymore).

Response: Yes, there are two identical copies of *PsAvr3c* in close proximity. Our primers used in Fig.1a can amplify both copies, resulting in a product of 459 bp. However, the PCR products from two independent *PsAvr3c* knockout homozygous mutants show clear single bands (345 bp and 352 bp, respectively) and no wild type band can be observed, indicating that both copies are simultaneously knocked out. Meanwhile, sequencing of PCR products suggest both copies are knocked out in an identical pattern. This could be due to high frequency of gene conversion rate in *P. sojae* (Chamnanpant et al. PNAS 2004).

In the revised manuscript, we use RT-PCR to demonstrate that no natural *PsAvr3c* transcript is detectable, demonstrating that both copies of *PsAvr3c* are impaired (Supplementary Fig. 2e). We add additional descriptions accordingly in lines 161-162.

Comments: Furthermore, the virulence data (loss of avirulence, fig 1b) are not convincing, as the phenotypes are not clear in such small panels. Bigger panels and quantification of growth would be desirable if the authors wish to make the point that Rps3c recognises Avr3c. For the purpose of this study, I am not sure whether the avirulence

data is needed (unless a connection is made between Rps3c function and alternative splicing, see below).

Response: In the revised manuscript, we provide a larger high-resolution image in Supplementary Fig. 3b. Meanwhile, we also add lesion length analysis (Supplementary Fig. 3c). We agree that the avirulence function of PsAvr3c is not the main topic in this manuscript. The reason we included the avirulence phenotype data is to support the conclusion that *PsAvr3c* was successfully knocked-out. We already removed this data from main figure to Supplementary Fig. 3.

Comments: Rather, a complementation assay (restoration of *P. sojae* mutants' virulence on roots expressing Avr3c) is more useful in the context of the main message as it would help demonstrate that Avr3c is the only virulence factor that has been disrupted.

Response: Genetic complementation assay will certainly strengthen the virulence role of PsAvr3c during infection. However, due to limitation of selection marker, generating stable gain-of-function *P. sojae* transformants on KO mutants is very challenging. Additionally, the suggested experiment is technically hard to achieve because bioassay of *P. sojae* effector mutants on transformed hairy roots is variable and unreliable, based on past experience from Prof. Yuanchao Wang's lab (co-author of this manuscript). It is likely that no clear or consistent phenotypes will be distinguished among the treatments. Furthermore, we cannot observe the oospores under the microscope as we demonstrated in Fig. 1d, because our *P. sojae PsAvr3c* mutants do not have the RFP marker. The contribution of PsAvr3c to phenotypic virulence is not a main claim of this manuscript. We hope the reviewer will agree that our current manuscript is acceptable without the complementation data.

Comments: Figure 1e & 5b needs modification. Oospores need to be identifiable by the reader (arrowheads and bigger panel?).

Response: We provide high-resolution and zoom-in images, and point out oospores with arrows in revised manuscript.

Comments: The authors identify GmSKRP1/2 as candidate host targets for Avr3c. Over-expression promotes virulence whereas silencing enhances immunity. Based on these observations, the authors state that GmSKRP1/2 are susceptibility factors. I disagree with this statement. To me GmSKRP1/2 appear to act as negative regulators of immunity. Naming them as such is more appropriate, given the subsequent claim that by targeting these proteins, immunity is subverted. In my view, susceptibility factors are proteins that aid pathogen infection in a way that is independent from immunity (e.g. increase nutrient availability).

Response: We accept the reviewer's suggestion, and we replaced susceptibility factor with negative regulator of immunity in the revised manuscript.

Comments: The authors state that GmSKRP1-spliceosome association is required for susceptibility. By saying this, the authors imply that this is connected to the activity of AVR3c. However, for this statement to stand up to scrutiny, the authors should demonstrate that AVR3c can still interact with GmSKRP1. If it still does, the authors statement is likely to be true, if not, the alternative hypothesis is that Avr3c-GmSKRP1 interaction is required for virulence. Alternatively, the statement should be modified such that it conveys the message that regulation of immunity requires this association.

Response: We appreciate reviewer for raising this issue, it would be good to determine whether PsAvr3c can interact with GmSKRPs under conditions of GmSKRP-spliceosome association or non-association status. However, to test this interaction in the soybean hairy root transient expression system is technically impossible. Instead, we select the alternative suggestion offered by the reviewer, and we re-word our statement in the revised manuscript accordingly and discuss the hypothesis in our discussion.

Comments: The authors conclude their work by demonstrating that GmSKRP1 has a role in alternative splicing by identifying genes that are more or less alternatively spliced in the presence of GmSKRP1. The authors then go on

to show that during infection and upon over expression of Avr3c, these alternative splicing events do take place. In addition, AS is firmly connected to AVR3c function as Avr3c mutants do not appear able to trigger these AS events during infection. Given the title of this MS and the claims made, I feel that these are the standout result that warrants expansion, which in turn, would justify the claim made. For example, a more comprehensive study using the AS data could be embarked on by: (i) Assessing whether there is significant enrichment of defense-associated genes amongst AS targets or perhaps in those AS events that result in inactivation of a given protein, (ii) demonstrate that GmSKRP1 and GmSKRP2 indeed interact with the target genes of interest or alternatively, assess whether the presence of Avr3c changes GmSKRP1/2 specificity towards their mRNA targets (which one may expect based on the observation that co-expression of Avr3c and GmSKRP1/2 gives an additional boost to infection. (iii) functionally validate the role of NAC-TF and/or WRKY transcription factors in immunity to *P. sojae*. The latter is a high-risk experiment however, given the number of possible candidate targets that could underpin the phenotype. AS of >1,000 could mean that the enhanced susceptibility phenotype is due to multiple targets genes or only a few major players. In either scenario, the chances of identifying a robust phenotype that connects GmSKRP1/2 function to a target, its fate and susceptibility is small. It is for that reason that the selection of two examples from >1,000 candidates is subjective and too suggestive.

Response: In the original manuscript, we concede that the RNA-seq analysis and experiments to verify AS events could be stronger. We made an effort to strengthen this part in the revised manuscript based on suggestions from both reviewer 1 and reviewer 3.

1) We sequenced more replicates (all the RNA-seq data now have three biological replicates) and performed a deeper AS analysis. Because the current knowledge on soybean defense genes is largely incomplete, many defense associated genes remain uncharacterized. We performed a GO analysis to detect enrichment but failed to observe significant enrichment of defense related genes. This is probably because after we harvested three biological replicates, the current number of genes with AS difference at high confidence narrowed down to 400. For such limited number of genes, the GO analysis may not work well.

2) Currently, our lab cannot perform an experiment to determine whether GmSKRPs targets specific transcripts *in vivo*. We believe this lies beyond the reasonable scope of expectations at this stage, but we agree this is an important issue to address in the future.

3) To address reviewer's concern, we produced overexpression and silencing soybean hairy roots. We firstly over-expressed specific *NAC* or *WRKY* transcription factors in soybean hairy roots, and test their effects on immunity to *P. sojae*. Ectopic expression lines are more resistant to *Phytophthora*. We added new data in the revised manuscript (Supplementary Fig. 17 c-e). Meanwhile, we also obtained the reduced resistance phenotypes (three replicates) from *NAC* and *WRKY* silencing hairy roots. However, it is impossible to guarantee silencing specificity, because of the presence of homologous genes with high sequence similarity in the soybean genome. *NAC* homologous gene Glyma.14G189300 that is not under AS is also silenced in the *NAC* silenced roots. The same situation occurs for *WRKY* genes (please see the following data). Although phenotypes are encouraging, we can not discriminate specific *NAC* and *WRKY* gene functions from silencing data.

We also added more experimental data on the AS analysis as reviewer 3 suggested. We tested more candidates by RT-PCR. Now, we had validated 10 AS events. Furthermore, we then test AS events from overexpressing soybean lines (GFP, GmSKRP1, GmSKRP1^{mut}, PsAvr3c, PsAvr3c^{M4}) and *P. sojae* infected soybean tissues (three wild type strains and PsAvr3c KO mutants). Please check new Fig. 7 and Fig. 8 in the revised manuscript.

Comments: The authors present numerous pieces of data for which I either question relevance, do not see a connection to the model or is open for alternative interpretation. For example, the authors claim that Avr3c over-expression re-locates GmSKRP1/2 to the nucleolus. Firstly, re-location suggests that protein present in the nucleoplasm is moved to the nucleolus, which is not correct as the nucleolar pool could be newly synthesized protein, Avr3c could prevent GmSKRP1/2 export from the nucleolus to the nucleoplasm and so on. Secondly, a connection is not evident between a change in localisation and GmSKRP1/2 altered function. The model (fig 7) does not mention this. The same goes for the prevention of nuclear localisation with Nuclear export signals, mutated NLS sequences etc. it is an extensive analyses that does not add much to the major finding.

Response: Since PsAvr3c is an effector with a predicted NLS, we believe that the experimental results showing that it goes to the nucleus and that GmSKRPs also localize in nucleus are very relevant to the story. However, the mechanism of GmSKRP1/2 relocalization in the presence of PsAvr3c, and its association with plant immunity remains unclear. Therefore, we try to be very cautious to our statements here. We briefly mention the observations in the results section of the manuscript but cut out other irrelevant data and descriptions. Please check our current version of the manuscript.

Comments: The authors use *P. capsici* in a number of experiments to validate or confirm results. Although of great value (see point below), again this distracts the reader from the main message. In short, I thus recommend for the authors to simplify this work by cutting out the least relevant pieces of data and strengthen the major conclusion by adding work supporting the major claim.

Response: We agree to cut out some least relevant pieces of data to strengthen the major conclusion. So we removed some data on *P. capsici-N. benthamiana* work (SFig.4, SFig.11, SFig.18 in previous version) from supplementary. We hope the reviewer agrees that major claims are still well supported after these actions.

Comments: The discussion could be more focussed around the model and pose additional questions. For example, Avr3c mediated changes in AS are discussed, but the exact mechanisms are not explored. Avr3c could act as a factor that helps recruit or bind SFs to pre-mRNAs or alternatively, change GmSKRP1/2 specificity. I find the “prevention of degradation” option a possibility, but not the most likely scenario. Rather, I suspect that lower turnover is due to retention of GmSKRP1/2 to pre-mRNA-spliceosome complexes.

Response: We agree, and we modified our discussion according to the reviewer's suggestions.

Comments: The authors use Rps3c lines to demonstrate that Avr3c is knocked out and leads to evasion of ETI. If true, either GmSKRP1/2 are guarded of Rps3c or alternatively (and perhaps more likely) an alternative splicing event on one or more mRNAs leads to a signal that triggers Rps3c mediated immunity. To my surprise, the connection to Rps3c is not discussed at all. I believe this to be of interest as the activation of AS by *P. capsici* and the presence of Avr3c homologs in other *Phytophthora* species, raises the prospect that Rps3c could be (modified and) deployed successfully against other *Phytophthoras*. This should either be mentioned in the discussion or the authors may as well remove all the work with the other *Phytophthora* species mentioned (homologs, *P. capsici* etc).

Response: This is a fair point. Our preliminary data suggests that GmSKRP1/2 are likely guarded by Rps3c. In our original version we did not discuss this idea. In our revised manuscript we added a new paragraph (lines 602-609) to discuss the potential ways that PsAvr3c-GmSKRPs interactions can activate ETI.

Comments: The text contains a fair number of errors. A detailed edit and check is required. For example, Lines 227-230 refer to the wrong supplementary figure and there are grammatical errors that need correction.

Response: We edited the manuscript carefully, and corrected many errors including those pointed out by reviewers.

Comments: Supplementary figure 13a. There is a thin line running through the western image. If it is meant to indicate an image modification, they should be more explicitly separated (together with the CBB stained panel below it).

Response: Yes, it is an image modification. We have replaced this with another image.

Reviewer #2 (Remarks to the Author):

Comments for the Authors

In this interesting manuscript, Huang et al expand our knowledge of a largely unexplored field: host-pathogen interactions at the level of alternative splicing. The authors start with the characterization of *Phytophthora* avirulence effector PsAvr3c and investigate its role in the pathogenesis in soybean. They show that PsAvr3c is a nuclear protein that binds to two soybean paralogous uncharacterized proteins, named GmSKRP1 and GmSKRP2, thus leading to their stabilization. The authors address functions of these proteins and demonstrate that they are important for plant immunity. Remarkably, the authors show that these proteins interact with several RNA binding proteins including orthologs of well-characterized *Arabidopsis* splicing factors SR45, U2AF35 and U1-70K, suggesting a role for GmSKRP1 and GmSKRP2 in pre-mRNA splicing. To get further insights into GmSKRP1/2 functions, the authors analyzed splicing changes using RNA-seq of GFP-GmSKRP1 over-expression lines and a GFP expressing control. The analysis identifies changes in several basic types of alternative splicing events in multiple genes, including NAC and WRKY transcription factors potentially involved in pathogen response. Moreover, the authors show that overexpression of PsAvr3c effector or its mutant version corroborates changes in splicing efficiencies of NAC and WRKY genes.

Though the authors present many solid results that are interesting for the plant community, there are several issues I found problematic as follows:

Comments: I may have missed it, but RNA-seq analysis was performed on two independent GFP-GmSKRP1 over-expression lines, which represented two biological replicates, and one GFP control line (in one replicate). The use of replicates (also for control) is essential for detecting differential alternative splicing. There is a high variability among two GFP-GmSKRP1 replicates as reflected by low number of the overlapping AS events. As the level of variability among replicates is so high, more replicates both for GFP-GmSKRP1 and importantly for GFP control are needed to deliver reliable lists of affected genes and AS events.

Response: The reviewer is correct. This is also a major concern of ours; how to identify the bona fide AS events in sequence data with a high level of variability. We concede that ectopic expression of transgenes in soybean hairy roots is variable and less consistent than many stably transformed plant systems, with regard to transgene expression and protein accumulation. To address these uncertainties, we sequenced an additional GmSKRP1 expressing sample and more GFP samples to ensure all the RNA-seq data have three biology replicates. Importantly, we used RT-PCR to validate more AS events in a more sophisticated manner (Fig.7).

Comments: The authors constructed a soybean transcript dataset with Cufflinks that was used as an input for rMATS to quantify differential alternative splicing. Currently, accurate reconstruction of transcript isoforms from Illumina RNA-seq short reads remains a significant challenge. For example, Steijger et al (Assessment of transcript reconstruction methods for RNA-seq. Nature Methods 2013;10(12) 1177-84) demonstrate that Cufflinks fails to recognize 40-60% of exons leading to mis-assembled and missing transcript isoforms thus resulting in poor quantification of alternative splicing. Therefore, thorough experimental validation is required to evaluate and support RNA-seq analysis.

Response: We thank the reviewer for pointing this out. Reconstruction of transcript isoforms is indeed a significant challenge in the RNA splicing field. We performed RNA-seq analysis to a higher standard and based the AS analysis on a more robust data set in the revised manuscript (Fig.6). Furthermore, we validated nine AS events by semi quantitative RT-PCR experiment (Fig.7) and three events by qRT-PCR (Fig.8) in the revised manuscript. We modified our discussion of this work and now cite Steijger's paper.

Comments: The authors performed RT-qPCRs for two putative pathogen defense genes, NAC and WRKY, to validate AS changes detected by RNA-seq analysis. Firstly, this is a biased approach. Secondly, two genes are not sufficient to evaluate the validation rate. Validation set should be expanded and should include random genes/AS events with a broad range of AS changes.

Response: This concern is similar to those expressed by the other reviewers. We have addressed these concerns by performing a new RNA-seq analysis, combined with more biological replicates. We validated 10 AS events by RT-PCR experiment (Fig.7) and three events by qRT-PCR (Fig.8) in revised manuscript. We hope these efforts will satisfy the reviewer's concerns.

Comments: Throughout the paper, the authors draw a link to Ser/Arg-rich (SR) proteins, based on the fact that GmSKRP1 and GmSKRP2 are rich in serines, lysines and arginines and that they interact with SR-like protein SR45. Bona fide SR proteins have characteristic features such as one or two RRM's at the N-terminus followed by a region rich in RS dipeptides. The precise definition and the unified nomenclature for plant and animal SR proteins have been established by the splicing community to avoid confusion and are provided for plants in Barta A, Kalyna M, Reddy AS. Implementing a rational and consistent nomenclature for serine/arginine-rich protein splicing factors (SR proteins) in plants. Plant Cell. 2010 Sep;22(9):2926-9 and for human in Manley J.L., Krainer A.R. (2010). A rational nomenclature for serine/arginine-rich protein splicing factors (SR proteins). Genes Dev. 24: 1073–1074. According to the established consensus SR45 is an SR-like protein and does not belong to SR proteins.

In addition, GmRBPs and NrRBP (also described as SR proteins – e.g. Supplementary Figure 16) do not belong SR family and represent SR-like proteins (potentially, homologs of SR45a/tra-2 as I can judge from a quick BLAST search). Moreover, GmSKRP1 and GmSKRP2 cannot be considered as “unconventional SR proteins” (line 125) as they do not have any features of SR or SR-like proteins. Therefore, all statements regarding SR proteins have to be revised and removed. For example, introduction (78-82, 125), results (348-351, 358, 376-378), discussion (543-546), Supplementary Figures 16 and 17b. List of references also needs a revision in this regard. To avoid any further confusion in the field, I would suggest renaming GmSKRP1 and GmSKRP2 to GmSKRP1 and GmSKRP1 as they are richer in lysines than arginines.

Response: We appreciated reviewer's comments and clear explanations. We removed SR statements and renamed the GmSKRP1/2 into GmSKRP1/2. Modifications are made accordingly throughout the revised manuscript.

The description of RNA-seq results and of the effect of GmSKRP1 on differential splicing is ambiguous, or at least unclear. Terms “genes”, “transcripts”, “mRNA transcripts”, “pre-mRNA transcripts” are used interchangeably in abstract and in the text (e.g. Abstract 34-35, Introduction 128 “more than one thousand mRNA transcripts”; Results 402 “>1,000 pre-mRNA transcripts”; Discussion 497 “>1,000 soybean genes” and 506 “Among the >1,000 genes identified”). Based on data in the Supplementary Table 2 I think that the authors have data on >1,000 AS events. The table contains 1108 entries, each corresponding to an AS event. There are 922 genes in this table, among them 142 genes have more than one AS event. From the data presented, it is not clear how many AS transcripts are generated, because an AS transcript can contain more than one AS event.

Response: We formatted these words as suggested by reviewer. Both events and genes are calculated based on three replicates for each case. In the revised manuscript, we demonstrated the number of genes throughout the manuscript and clarified the number of events when necessary.

Other points/suggestions:

Comments: 64-65 “RNA splicing edits pre-mRNA transcripts by removing introns from nascent mRNA transcripts and joining exons together” and other similar sentences in the text - mRNA (messenger RNA) is a spliced transcript, introns are removed from pre-mRNA.

Response: Thank you for pointing this out. We changed nascent mRNA to nascent RNA, and corrected other similar words.

Comments: 66 “This process (splicing) takes place within the nucleus either co-transcriptionally or immediately after transcription.” – considerable delay in splicing is widespread, e.g. in intron retention transcripts (Boothby 2013, Wong 2013, Shalgi 2014).

Response: We rephrased the sentence “Evidence from yeast and mammals suggests splicing happens co-transcriptionally for the majority of genes while post transcriptionally for many individual genes” in lines 66-68.

Comments: 67-69 “Alternative splicing (AS) generates multiple transcripts from a single gene and can result from intron retention, exon skipping, or selection of an alternative 5’ donor sites or 3’ acceptor site” – AS is not limited to these events. In addition to mutually exclusive exons, which also represent basic AS events, there are non-canonical AS events such as microexons and exons (Irimia et al. Cell. 2014 Dec 18;159(7):1511-23 and Marquez et al. Genome Res. 2015 Jul;25(7):995-1007)

Response: We cite the suggested papers and modified our statement accordingly.

Comments: 69-70 references 7 and 8 - replace with the newer, less specific reference, for example Reddy et al. Complexity of the alternative splicing landscape in plants. Plant Cell. 2013 Oct;25(10):3657-83. doi: 10.1105/tpc.113.117523

Response: We replaced the reference and modified our statement accordingly.

Comments: 72 wrong reference for Arabidopsis – replace with Marquez et al. Genome Res. 2012 Jun;22(6):1184-95

Response: We cited the suggested papers and modified our statement accordingly.

Comments: 93-94 “Another example is provided by the RCT1 gene of Colletotrichum trifolii, which confers resistance against multiple races of fungal pathogen” - it is a Medicago truncatula gene, which confers resistance to Colletotrichum

Response: Thank you for pointing this out. We corrected the statement.

Comments: 102 reference 15 – replace with the reference to the original paper Yang et al Proc Natl Acad Sci U S A. 2008 Aug 26; 105(34): 12164–12169.

Response: We cite the paper accordingly.

Comments: 139 “The qRT-PCR data demonstrated that the expression of PsAvr3c is induced at early stages of infection and is greatest at 1 hpi, which is consistent with RNA-seq data (Supplementary Fig. 1a, 1b)” - I may have missed it, but I did not find this RNA-seq analysis (also no description in Methods)

Response: We added descriptions of RNA-seq analysis details in Methods. RNA-seq data is from Jing M et al. Nat Commun. 2016; we cite the paper accordingly.

Comments: 217 Sequences of GmSKRP1/2 (and NbSKRP) should be added to Supplementary Table 3, which contains sequences of all proteins mentioned in the manuscript except these.

Response: We added the sequence as suggested. Please check new Supplementary Table 3.

Comments: 219 “GmSKRP1 and GmSKRP2 are 558 bp and 552 bp in length, respectively” – these are the lengths of the coding regions of these genes. The genes themselves are longer as they contain introns and UTRs.

Response: The numbers refer to CDS, we clarified this in our revised manuscript in lines 224-226.

Comments: 258 “We observed that RFP-GmSKRP1/2 fusion proteins predominantly accumulated in the nucleoplasm, with a low proportion of nuclear speckle localization.” – I do not see any speckles.

Response: We added speckle localization data and statistical data in the revised supplementary Fig. 8a, 8b. And add descriptions in results part in lines 263-264.

Comments: 432 “alternative splicing of the two predicted transcriptional factor genes in the presence of PsAvr3c effector results in emergence of premature stop codons in retained intron sequences. Thus, the transcripts are likely be degraded by non-sense mediated RNA decay or translated as truncated nonfunctional proteins” 524-526 – recent evidence implicates that intron retention transcripts are not degraded by nonsense-mediated mRNA decay (Kalyna et al. Nucleic Acids Res. 2012 Mar;40(6):2454-69, Leviatan et al. PLoS ONE 8, e66511 (2013) and papers from human/mammalian field) as they are retained in the nucleus and therefore are not accessible to NMD machinery (Gohring et al. Plant Cell. 2014 Feb;26(2):754-64 and other papers, also from human/mammalian field – Boothby et al 2013, Wong et al 2013, Shalgi et al 2014).

Response: We modified our statements accordingly and cite the suggested paper.

Comments: 455 “Genes encoding SKRP-like proteins are prevalent in plants” – 1) no data is presented to support this statement; 2) are SKRPs plant-specific?

Response: Yes, SKRP-like protein homologs can be found in both dicot and monocot plants, but this is still under investigation and the full phylogenetic analysis of this protein family is not complete. At this stage, we are able to

illustrate selected SKRP homologs from a few plant species (Supplementary Fig. 19a). We also added to the discussion related to SKRP prevalence in other organisms in revised manuscript, as suggested.

Comments: 459 “pathogen effector interferes with host RNA splicing machinery to reprogram the processing mRNA transcripts, and we propose that pathogens suppress host immunity at mRNA modification layer.” – 1) to reprogram the splicing of pre-mRNA (mRNA transcripts are spliced products); 2) RNA modification – this term is mostly applied for changes to the chemical composition of RNA (capping, m6A, m5C, pseudouridine etc), aka “epitranscriptome”.

Response: We rephrased the wording in revised manuscript as the reviewer suggests.

Comments: 467 “A KH type RBP protein” – it says the same thing twice: RBP protein = RNA binding protein protein

Response: We corrected in the revised manuscript.

Comments: 472 reference 33 is not complete

Response: Thank you for pointing this out. We corrected in the revised manuscript.

Comments: 513 “...mRNA sequencing is not sensitive enough to detect the changes in splicing for many genes, as other authors have speculated previously³⁴. Therefore, the actual number of affected splicing events could be greater than we identified in this initial analysis.” – the problem is not in the sensitivity of RNA-seq, which with sufficient sequencing depth can detect even minor unspliced pre-mRNAs, but in RNA-seq analyses based on assemblies of transcripts from short reads, miss-classification of complex AS events and on low sensitivity and precision of quantification programs.

Response: We appreciate the comment, and we have rephrased the wording accordingly.

Comments: 542 and throughout the text – I would suggest to avoid using superscript in U2AF³⁵ and to replace it with U2AF35, in order to avoid confusion with the references.

Response: Thank you for pointing this out. We corrected in the revised manuscript.

Methods:

Comments: Plant growth conditions are not described properly (different species, different experiments)

Response: Thank you for pointing this out. Growth conditions are described in more detail in revised manuscript.

Comments: No RNA-seq description for data presented in Supplementary Fig. 1.

Response: This is now described in revised manuscript.

Comments: RNA-seq paragraph is lacking essential descriptions, e.g. no description of RNA isolation, library preparation, RNA-seq read characteristics (length, single or paired end?), no information on soybean genome used for read mapping.

Response: We added details for the RNA-seq and data analysis part.

Figures:

Comments: Figure 7 is not mentioned in the text.

Response: We reorganized the figures, and verified this Fig is mentioned in the revised manuscript.

Comments: 1147 Fig. 7 legend “In particular, some defense related genes generate alternative splicing isoform and yield truncated nonfunctional proteins” – there is no evidence that they yield any proteins. In addition, truncated proteins can be functional.

Response: We agree, and we rephrased the words in Fig legend in the revised manuscript.

Reviewer #3 (Remarks to the Author):

The manuscript “An oomycete plant pathogen reprograms host pre-mRNA splicing to subvert immunity” by Huang et al. describes a functional characterization of the effector PsAvr3c from *Phytophthora sojae*, a soybean pathogen. In a large number of well-controlled and high-quality experiments, the authors show that PsAvr3c binds to GmSKRP1/2, which are serine/arginine/lysine rich proteins of unknown functions. The authors further demonstrate that this interaction results in GmSKRP stabilization and partial relocalization into the nucleolus. Using immune precipitation and mass spectrometry, interacting partners of SKRPs are identified, including several splicing factors. Finally, splicing patterns upon SKRP overexpression are analyzed in a transcriptome-wide manner. Two of the events within defense genes are independently verified; furthermore, some changes in these events are also found upon infection with *P. sojae* in an PsAvr3c effector-dependent manner.

This work provides a model of effector-mediated changes in alternative splicing as critical component of virulence function. This would be indeed a very interesting mechanism. The manuscript includes many interesting data, with the only major weakness in the analysis of the alternative splicing pattern changes. First, it would have been more interesting to analyze transcriptome-wide AS in plants infected with *P. sojae* with functional and non-functional Avr3c. Second, the number of validated events is very low, and any correlation between SKRP overexpression and the infected samples is not convincing. For details see below. I think the authors should extent some of these analyses, and clearly highlight the open questions. With these edits, I would see this work as an important contribution to the field.

Major comments:

Comments: 1) The splicing analyses in Fig. 6 show changes in the splicing ratios. These changes can result from alternative splicing or altered stability of one of the isoforms. Previous work had shown inactivation of NMD in plants upon pathogen infection. Accordingly, the changes observed in this study might result from NMD inactivation. In line with this, intron-retained transcripts with NMD target features over-accumulate upon GFP-SKRP1 expression. To provide evidence for changes in splicing, the authors need to separately quantify both splicing forms (intron-spliced and intron-retained); if changes occur on the level of splicing, opposite changes in the two isoforms are expected.

Response: It is a good question. From our previous data, we can not discriminate the alternative splicing or altered stability of any of the isoforms. In our revised manuscript, we use semi-quantitative RT-PCR to examine different isoforms (Fig. 7). As reviewers could see in most of the cases, the total signal intensities from two isoforms are similar, whereas the ratio (intron-spliced / intron-retained) that are changed in GFP-GmSKRP1 and GFP-PsAvr3c expression lines. The data is indeed consistent with reviewer’s speculations. We hope reviewer could agree with us that the current data is sufficient to support our conclusion.

Comments: 2) I was wondering why the authors haven’t performed RNA-Seq from soybean plants infected with the different *P. sojae* strains (analyzed in Fig. 6e). This is expected to result in more physiological data than the analysis of the overexpression lines, also given that redistribution of SKRP1 in an effector-dependent manner was observed.

Response: We agree that a direct comparison of RNA-seq data from infected soybeans (challenged with wild type and PsAvr3c KO *P. sojae* mutants) is a biologically appealing experiment. However, with due respect to the reviewer and editor, we would like to raise our own concerns with such an experiment and suggest an alternative.

Based on previous analyses, it is know that there are massive transcriptional changes during infection of soybean

by *P. sojae*, which will create a lot of noise. More importantly, *Phytophthora sojae* secretes several hundred cytoplasmic effectors besides PsAvr3c (Tyler et al. Science, 2006, Jiang et al., PNAS, 2009), some of which may also directly or indirectly affect splicing. So, although the suggested experiment might work, we think that it is quite risky because the signal (PsAvr3c effects) could be lost in the noise (infection related changes that are independent of PsAvr3c). In fact, we do not know of any study to date that has successfully used such an approach to characterize the functionality of an oomycete virulence effector. (two relative publications for reviewers and editor to consider are: Jing M et al. Nat Commun. 2016; Boevink PC et al. Nat Commun. 2016). We also note that reviewer 1 and reviewer 2 do not suggest such an experiment, despite that they also would like to see the RNA-seq analysis improved.

To strengthen our manuscript, we suggest that ectopic expression of PsAvr3c in hairy roots is a preferable experiment than the infection site analysis. Our feeling is that it will provide a cleaner analysis of the effect of PsAvr3c on host cellular operations, since it is not confounded by the massive effects of infection and host defense responses. Ectopic expression in the plant is a commonly used approach to probe effector functionality in other host-pathogen systems, and to isolate effector-specific effects from other infection related processes. We performed a deep RNA-seq analysis with treatments such as expression of GmSKRP1, PsAvr3c, and GFP, and provide three biological replicates for each. Then we verified those interesting AS events by RT-PCR experiments, on a larger scale than we previously did, as reviewers suggested. We then test 10 transcripts, identified as AS from the RNA-seq work, in *P. sojae* infected soybean tissues (three wild type strains and PsAvr3c KO mutants) and in overexpressing soybean lines (GmSKRP, GmSKRP^{mut}, PsAvr3c, PsAvr3c^{M4} and GFP).

Comments: Furthermore, Fig. 6e lacks a control with a strain carrying an unrelated effector, such as PsAvh52. Relative to this control, the WT and T71 are expected to result in a decrease of the splicing efficiency, while the knockouts should show splicing patterns similar to the control.

Response: The logic of this suggestion is correct, however, it is not easy to achieve.

All known *P. sojae* strains carry PsAvh52, so there is such a control as suggested by reviewer 3. We supposed an alternative is to test PsAvh52 KO mutants. Unfortunately however, we do not have PsAvh52 knock out mutants. *P. sojae* transformation and KO are not easy or routine. We put in a lot of effort to generate only a few PsAvr3c KO mutants. The point here is to demonstrate that PsAvr3c mutants have increased splicing efficiency. We have added more *P. sojae* natural strains (FJ8, AH14) and *PsAvr3c* in-frame CRISPR mutants. Please check new Fig. 7 and Fig. 8 in the revised manuscript.

Comments: In the absence of RNA-Seq data from these samples, it would be more convincing to analyze instead of only two some more candidates (at least 5 more randomly chosen candidates from the event list of the overexpression lines).

Response: After harvesting more biological replicates, we reanalyzed the RNA-seq data. In the revised manuscript, we validate a total of 10 AS candidates by RT-PCR analysis (revised Fig. 7). Three of them are further tested by qRT-PCR (revised Fig. 8a, 8b, 8c).

Comments: 3) I find the observation of SKRP1/2 relocation to the nucleolus in the presence of Avr3c interesting in light of a splicing regulatory effect. As a possible explanation, SKRP1/2-mediated splicing function might be suppressed in the presence of Avr3c by removing these factors from the sites of splicing (nucleoplasm and speckles). The authors should discuss this possibility and clearly state that splicing is not expected to occur in the nucleolus.

Response: This perspective is different from that of reviewer 1, who felt that this part is not relevant, and suggested to edit it out. We respectively agree with reviewer 3, who feels the localization work is relevant to the study of PsAvr3c and GmSKRPs functionality, so we would like to retain these results in the revised manuscript. Currently, our genetic data suggest overexpression of PsAvr3c and GmSKRP1 functions similarly. Therefore, our hypothesis is that GmSKRPs is a negative regulator of AS and PsAvr3c enhances GmSKRPs functionality rather

than suppressing its activity. The idea that PsAvr3c relocates GmSKRPs from nucleoplasm/speckles to nucleolus and suppresses splicing makes sense and can be supported by cell biology data, but it is conflict with our observation that GmSKRP1 overexpression (mainly localized in nucleoplasm) impairs normal splicing. Therefore, we are cautious about interpreting the cell biology results because we do not know the real mechanism of PsAvr3c effector action on GmSKRP1/2. We shorten the cell biology part in the revised manuscript in an attempt to meet these contrasting reviewer concerns.

Further comments:

Comments: 1) The soybean infection assay shown in Fig. 1b should be explained in more detail, in particular the interpretation of the results. The replicates should be included in Supplement. Could the spreading zone also be quantified as in Fig. 1c?

Response: According to reviewer1's suggestion, we removed Fig. 1b to Supplementary Fig. 3. We provide better infection images with more details and quantification data.

Comments: 2) Fig. 2d shows the results from isothermal titration calorimetry for measuring the binding affinity between PsAvr3c and GmSKRP1. The authors provide a Kd as a representative value of three independent experiments. It would be more convincing to show all three experiments and provide all three values or a mean Kd with standard deviation.

Response: We demonstrate Kd in mean value with standard deviation. The values have been corrected in the revised manuscript. (Three original measurement results are 3.38 μ M, 3.52 μ M and 3.14 μ M)

Comments: 3) Some of the quantitative data in Supplement need better description, always provide number of replicates, types of error bars, and displayed values (mean?).

Response: We provide more details on these quantitative data in revised manuscript.

Minor points:

Comments: 1) l. 249-251: "To determine whether greater accumulation of GmSKRP1/2 might be due to enhanced gene expression, qPCR was used to quantify GmSKRP1/2 transcriptional level." Statement should be changed to "transcript level" as steady state levels are measured, which can be influenced by the transcriptional rate and RNA turnover rate.

Response: We corrected this in revised manuscript.

Comments: 2) Fig. 7 is not mentioned in the text. Furthermore, some of the symbols like the ovals need to be defined. There is no evidence for translation of the alternative splicing products; the corresponding statement in the legend should be toned down or better removed.

Response: We reorganize the figures, and mention the model (new Fig 8d) in the revised manuscript. Symbols are more carefully defined. We agree with the reviewer about the wording, and we modified it accordingly.

Reviewers' comments:

Reviewer #1 (Remarks to the Author):

after having evaluated this MS, i believe that the authors have incorporated all of my suggested changes (that were reasonably achievable). i therefore think that the MS is suited for publication in this journal. additional comments are listed below:

1. Authors show that levels of GmSKRP1/2 increase in the presence of Avr3c and that this is due to stabilisation since on the transcriptional level, differences are absent. it may therefore be worth to assess whether GmSKRP1/2 is subject to AS, leading to more stable isoforms.
2. Figure 3b (right hand panel). the Avr3c mutant is smaller (left hand side panel) but in the right panel, size differences are minimal or absent. Please revisit figure and assess whether they were run to the same extent/time (if not, an additional size marker needs to be included to make this clear) or if there are any other issues (i think it is the former, the level of separation appears to be different).
3. Figure 4a and c. two images are shown for each treatment. please indicate the difference between each.
4. the authors say in the legend of fig 5: "GmSKRP1 associates with alternative spliceosome components in vivo but GmSKRP1mut does not". it is more appropriate to say that GmSKRP1 interacts with complex(es) that contain.... direct interaction between each of the components and GMSKRP1 has not been demonstrated. make sure that the MS text also correctly states this.
5. Figure 7. detection and validation of AS events. for some of the genes presented here (Cysteine synthase, Nac TF factor) there appears to be an additional band that is not marked or annotated (whilst it seems to be linked to Avr3c activity). it is also not clear how relative abundances were calculated (as they do not add up to 1).
6. Figure 8. i like the model though i wonder if it could be made a little more specific. i would suggest to augment "immunity" in the model figure with some of the biological processes that may be affected by Avr3c induced AS. e.g, defense gene induction, etc. this is optional though. i also did not immediately get that the thickness of the red arrow represented the amplitude of the response. perhaps this point could be made in a different way? same can be said for stabilisation of GMSKRPs, a line showing inhibition of this process may be more explicit and clear? Finally, what data does the line showing inhibition of AS (in the presence of Avr3c) represent? if there is not any, this should be removed.

Reviewer #3 (Remarks to the Author):

Huang et al. have provided a revised version of their manuscript "An oomycete plant pathogen reprograms host pre-mRNA splicing to subvert immunity". The authors have extended the splicing

analyses and added further data to support their claims. While I see some improvement in the manuscript and still consider this work of high relevance, several major concerns remain. Details are listed below, but one major issue is still the splicing analysis. The authors have used for validation co-amplification of splicing variants, followed by quantification of splicing variant ratios (Fig. 7). This assay is widely used and accepted for analyzing splicing variant ratios; however, quantitative statements for single bands cannot be made. The authors make such statements based on intensities of single bands at several places in the manuscript and in their rebuttal, to substantiate that the changes in the splicing variant ratios occur on the level of splicing and not transcript stability. Based on several previous publications, pathogen infection causes NMD inhibition. Accordingly, the changes in the splicing ratios in the current work might result from stabilization of the intron-retained forms, and not from a change in alternative splicing. As I suggested before, it would be straight-forward to distinguish between these possibilities by measuring levels of the individual splicing variants via qPCR. Reciprocal changes would be expected in case of a change in alternative splicing. In the absence of a molecular characterization of SKRP1 functioning, I think excluding specific stabilization of the intron-retained transcript is a key experiment to support the authors' claim of changes in splicing. Furthermore, I had suggested performing the RNA-Seq with plants infected with *P. sojae* carrying functional or non-functional PsAvr3c. The authors are concerned that due to the presence of many additional effectors, no specific change might be visible. However, isn't that in contrast to the data in Figs. 7 and 8, where clear differences can be seen? I don't think that including additional RNA-seq is absolutely required, however, the authors should discuss their experimental design and future directions.

Further major issues:

1) The validation of AS events has been extended and included as Figure 7. Most of the data look convincing, but the following modifications should be made. First, the legends says "...gene expression levels were analyzed...". However, this method does not allow quantifying gene expression as normalization to a reference transcript is missing and as the method is not quantitative. Only AS ratios can be deduced from this data. The last two candidates are not meaningful from my point of view. Glyma.19G081300 gives only one band. If I get the model right, exon2 can be skipped or excluded. The reverse primer cannot bind to the variant lacking exon2, therefore co-amplification for ratio analysis cannot be used. Instead, qPCR for analyzing the level of the variant with exon2 relative to total expression (or to the exon3 variant, in case these are mutually exclusive exons) of this gene would be needed to support a switch in splicing variant ratios. Glyma.16G073600 gives only one band, and is included as a control that AS efficiency is not altered. How can this serve as a control for AS if no alternative splicing is visible in any sample? Has AS for this gene been reported before, or why has it been chosen? Statements in the text such as "...total level from two isoforms remains similar" (l. 418-419) and "...does not yield reduction of intron-spliced signals..." (l. 422) lack experimental support, as this method cannot be used to quantify single transcripts or total expression level.

2) Figure 6 contains a summary of the RNA-seq data with respect to the alternative splicing changes. Fig. 6a gives a summary of genes showing significant AS changes in the different samples. Separate Venn diagrams have been used for the four basic AS types. It would be more logical to display events here, and maybe provide corresponding gene numbers in parenthesis, for the following reason. One gene

might have for example an IR event in one sample, and an ES event in another sample. There is no overlap in the AS event category, but it would give an overlap for the gene. I am not sure how the authors have dealt with this kind of candidates. Furthermore, I am not sure if I interpret the data display in the Circos diagram correctly. What does “heat map view of gene density including differential alternative splicing...” (from corresponding legend) mean? Does every colored line indicate the position of a gene with an AS event, and does the line color somehow indicate how close genes with AS events are? Finally, the red lines are supposed to connect the position of the GmSKRP1 genes with genes containing SKRP1/Avr3c-regulated events. What can be concluded from this? I think it would be sufficient to indicate the positions of the AS events, genes, and GC content, and move this diagram to Supplement, as it is purely descriptive.

3) For several figures, it remains unclear what kind of replicates have been used. For example, legend to Fig. 4c states “Three independent experiments showed similar results”. Legend to Fig. 4d says “Statistics of (c). Means and standard errors from replicates are shown...” and in the diagram it says $n = 10$. If I understand it correctly, three independent experiments have been performed, and 10 replicates from one experiment have been analyzed. Another example is legend to SFig. 18: “Experiments were repeated three times with similar results” In the graphs, it says $n = 3$. Are these technical replicates or biological replicates? Thus, a clearer and more detailed description would be needed in several legends. Some legends still do not contain any information on number of replicates and types of error bars, e.g. SFig. 1a, c, d.

4) It is difficult to see the details of the gene models displayed in Fig. 8, in particular arrows designating primers and the symbols for premature termination are too small, and colors of arrows are indistinguishable on a print out. The display could be reduced to the relevant part of the gene model to increase size of the relevant regions. Furthermore, the display of the qRT-PCR data includes mutants of GmSKRP1 (mut) and PsAvr3c (M4). These data are not mentioned in the text. Based on my expectation that the wild type but not the mutant versions alter splicing efficiency, I think that the labels for “PsAvr3c” and “GmSKRP1mut” are swapped. The values are defined as “...splicing efficiencies (the ratio spliced RNA over unspliced)...” (l. 440-441). I would expect that splicing efficiency represents the ratio $\text{spliced}/(\text{spliced} + \text{unspliced})$. Given that some values are clearly above 1, and as 1 would be the maximum, i.e. 100% spliced, I guess that the values were indeed calculated differently. The quantification would require a better description.

5) The authors express two of the splicing targets, Glyma.02G222300 and Glyma.03G220800, in hairy roots, which results in increased resistance to *P. sojae* (SFig. 17c-e). It is unclear what conclusion can be drawn from this experiment in the context of the splicing phenotype. First, are the genes overexpressed in the hairy roots? Transcript levels should be analyzed compared to the GFP control line. The authors show in their other experiments that the splicing change does not involve changes in total transcript levels. So how could the phenotype relate to changes in alternative splicing? Second, were genomic or cDNA sequences used for these constructs, and can alternative splicing still occur?

6) The manuscript still contains many grammatical errors. Usage of the terms mRNA/pre-mRNA is still not correct at several places, e.g. l. 75 “...over 60% of mRNA transcripts ...undergo AS”. Several

statements in introduction need clarification: 1. l. 70-71 "...mutually exclusive exons observations show that IR is a frequent event in both in human and in plants." - meaning unclear. 2. l. 81 "...many non-snRNPs are also involved in spliceosome operations." - Which ones? 3. l. 124 "...GmSKRPs, novel components of plant spliceosome." - referring to these proteins as interactors of spliceosomal components might be more appropriate as long as their molecular function is unclear.

Further comments:

- 1) Legends: Most legends summarize the findings, sometimes even twice; e.g. legend to Fig. 2c: "PsAvr3c interacts with GmSKRP1/2 ... confirms that FLAG-PsAvr3c specifically interacts with GFP-GmSKRP1/2..."; or legend to Fig. 2d: "Measuring binding affinity between PsAvr3c and GmSKRP1 ... binding affinity of PsAvr3c to GmSKRP1 was determined....". The legends could be reduced to a description of the figures, but should also sometimes provide more details. For example, for none of the immunoblots it is mentioned that labels with numbers represent molecular weight in kDa. This kind of information should be indicated in the figure or legend.
- 2) Legend to SFig. 11: legend title incomplete; change to "...in soybean immunity against *P. sojae*"
- 3) Legend to Supplemental Fig. 18: "... transcriptional level of NAC..." should be replaced by "... transcript level of NAC..." as these qPCR analyses measure steady state levels, i.e. sum of transcription and decay, and not transcription itself.

In the revised manuscript we addressed the reviewers comments and incorporated their suggestions, which further strengthened our manuscript. Below, we have provided point-by-point response to reviewers' comments.

Reviewer #1 (Remarks to the Author):

after having evaluated this MS, i believe that the authors have incorporated all of my suggested changes (that were reasonably achievable). i therefore think that the MS is suited for publication in this journal. additional comments are listed below:

Comments: 1. Authors show that levels of GmSKRP1/2 increase in the presence of Avr3c and that this is due to stabilisation since on the transcriptional level, differences are absent. it may therefore be worth to assess whether GmSKRP1/2 is subject to AS, leading to more stable isoforms.

Response: This is an interesting point that we addressed in the revised version. We looked through the RNA-seq data and did not find significant AS events for *GmSKRP1/2* in PsAvr3c overexpressed hairy roots. In addition, we did not observe additional GmSKRP protein bands in the presence of PsAvr3c in our western blot data (see Fig.3). Therefore, we speculate that GmSKRP1/2 have no detectable AS isoform, which could explain the increased stability of GmSKRPs. We added this in the discussion section of the revised manuscript.

Comments: 2. Figure 3b (right hand panel). the Avr3c mutant is smaller (left hand side panel) but in the right panel, size differences are minimal or absent. Please revisit figure and assess whether they were run to the same extent/time (if not, an additional size marker needs to be included to make this clear) or if there are any other issues (i think it is the former, the level of separation appears to be different).

Response: Yes, the protein gel separation is different. To avoid confusion, we replaced Fig 3b left panel with a new image in the revised manuscript.

Comments: 3. Figure 4a and c. two images are shown for each treatment. please indicate the difference between each.

Response: Thank you for pointing this out. We reorganized these figures and added additional description in revised figures.

Comments: 4. the authors say in the legend of fig 5: "GmSKRP1 associates with alternative

spliceosome components in vivo but GmSKRP1mut does not". it is more appropriate to say that GmSKRP1 interacts with complex(es) that contain.... direct interaction between each of the components and GMSKRP1 has not been demonstrated. make sure that the MS text also correctly states this.

Response: Thank you for this suggestion. We corrected this in the revised manuscript.

Comments: 5. Figure 7. detection and validation of AS events. for some of the genes presented here (Cysteine synthase, Nac TF factor) there appears to be an additional band that is not marked or annotated (whilst it seems to be linked to Avr3c activity).

Response: We agree that there are additional bands in some genes. To be honest, we have no idea about these bands. They are likely other unpredicted isoforms or unspecific PCR bands. However, as reviewer mentioned they seems to be linked to PsAvr3c activity. Although some unexpected bands appeared, AS of these genes are clearly affected in a PsAvr3c dependent manner. Our semi-quantitative RT-PCR data supports our conclusion.

Comments: it is also not clear how relative abundances were calculated (as they do not add up to 1).

Response: An equal amount of cDNA template in each reaction was verified by amplifying soybean actin gene CYP2, and then PCR reaction with primers specific to each gene was conducted. The intensities of PCR products were quantified by ImageJ software, and the relative abundance were then calculated by using bottom (or top) bands in the left lane as reference (set to 1). We added additional descriptions in the figure legend.

Comments: 6. Figure 8. i like the model though i wonder if it could be made a little more specific. i would suggest to augment "immunity" in the model figure with some of the biological processes that may be affected by Avr3c induced AS. e.g, defense gene induction, etc. this is optional though. i also did not immediately get that the thickness of the red arrow represented the amplitude of the response. perhaps this point could be made in a different way? same can be said for stabilisation of GMSKRPs, a line showing inhibition of this process may be more explicit and clear? Finally, what data does the line showing inhibition of AS (in the presence of Avr3c) represent? if there is not any, this should be removed.

Response: We reorganized the model in the revised manuscript. As suggested by this reviewer, we revised the model. We removed "red arrow" and the "inhibition line of AS". Given the evidence that PsAvr3c suppression of immunity is clear and important for this manuscript, we modified the figure to depict reduced plant immunity.

Reviewer #3 (Remarks to the Author):

Comments: Huang et al. have provided a revised version of their manuscript "An oomycete plant pathogen reprograms host pre-mRNA splicing to subvert immunity". The authors have extended the splicing analyses and added further data to support their claims. While I see some improvement in the manuscript and still consider this work of high relevance, several major concerns remain. Details are listed below, but one major issue is still the splicing

analysis. The authors have used for validation co-amplification of splicing variants, followed by quantification of splicing variant ratios (Fig. 7). This assay is widely used and accepted for analyzing splicing variant ratios; however, quantitative statements for single bands cannot be made. The authors make such statements based on intensities of single bands at several places in the manuscript and in their rebuttal, to substantiate that the changes in the splicing variant ratios occur on the level of splicing and not transcript stability. Based on several previous publications, pathogen infection causes NMD inhibition. Accordingly, the changes in the splicing ratios in the current work might result from stabilization of the intron-retained forms, and not from a change in alternative splicing. As I suggested before, it would be straight-forward to distinguish between these possibilities by measuring levels of the individual splicing variants via qPCR. Reciprocal changes would be expected in case of a change in alternative splicing. In the absence of a molecular characterization of SKRP1 functioning, I think excluding specific stabilization of the intron-retained transcript is a key experiment to support the authors' claim of changes in splicing.

Response: We appreciate the concern of this reviewer and agree that the differences in isoform levels could be due to NMD. Our responses are listed as the following:

(1) We appreciate the reviewer's comments that providing qPCR data of each isoform would be more accurate. However, we believe our current data is sufficient to support our claim and qPCR data won't change our conclusion that isoform level are altered. For many examples we showed in Figure 7 (01G238400, 02G150800, 03G220800 etc), comparing over-expression line with the mutant, there are reciprocal changes among the two isoforms (one isoform increased while other isoform decreased), this is consistent with changes in alternatively splicing. As this reviewer pointed out co-amplification of splicing variants by semi-quantitative PCR followed by quantification of splicing variant is widely recognized by scientists working on AS from animal and plant research community (Here we listed two recent publications: Ajiro M, et al. *Nucleic acids research*, 2016 and Haussmann IU, et al. *Nature*, 2016). Please also note our quantification is not a direct comparison of spliced/unspliced forms, it is a relative quantification of each isoform in one gel. In this case, data in Fig. 7 not only reflect ratio of splice isoforms, but also tell us the abundance (expression) of each isoform. In previously publications, it is clearly mentioned that it is used for analysis of gene expression and splicing. (for example: "Plant serine/arginine-rich proteins: roles in precursor messenger RNA splicing, plant development, and stress responses" *Interdisciplin Review RNA*, 2011. Fig. 2: Analysis of expression and splicing of Arabidopsis serine/arginine-rich (SR) genes in root, stem, leaf, inflorescence, and pollen...)

(2) Furthermore, reviewer 3 seems to more focused on intron-retained isoforms. However, our AS analyses data demonstrated that there are other types of AS. In revised manuscript, we added two clear exon skip examples. Exon skipping normally do not generate AS isoforms with a premature stop codon which is important for NMD recognition and activation. In addition, many of these AS events overlaps with AS that are previously documented (Shen Y, et al. *Plant Cell*, 2014). So effector triggered AS change should be supported by our data, considering all of the work presented.

(3) We appreciate the reviewer to raise the possibility of NMD. In some cases (Fig.7 03G099900), we did not observe an obvious change in the spliced isoform while an increase in the intron-retained isoform was observed. This is indeed consistent with inactivated NMD while in some other cases, the changed alternative splicing as well (e.g. increased transcription in addition

to decreased splicing efficiency), we believe this is hard to distinguish even with qPCR quantification of each isoform. For this reason, we have now carefully discussed potential role of NMD in our revised manuscript. In the meanwhile, based on the reason we listed in (1) (2) and elsewhere in this study, our data strongly suggest alternative splicing is regulated by PsAvr3c.

(4) According to reviewer's suggestion, we performed experiment to validate one of the AS events. Please see the following data, we measured the transcript levels of the individual splicing variants of NAC transcriptional factor (Glyma.02g222300) via qPCR, taking CYP2 gene as internal reference. The result shown there are reciprocal changes among the two isoforms, we believe this data support our claim of changes in splicing.

Furthermore, I had suggested performing the RNA-Seq with plants infected with *P. sojae* carrying functional or non-functional PsAvr3c. The authors are concerned that due to the presence of many additional effectors, no specific change might be visible. However, isn't that in contrast to the data in Figs. 7 and 8, where clear differences can be seen? I don't think that including additional RNA-seq is absolutely required, however, the authors should discuss their experimental design and future directions.

Response: We agree that a direct comparison of RNA-seq data from infected soybeans is a biologically appealing experiment. However, as this reviewer pointed out that such an experiment is not absolutely required. Although we didn't do RNA-seq on infected tissue, we validate several AS events by RT-PCR and qRT-PCR on infected soybean tissue and compare to wide type and two PsAvr3c KO mutants in Figs. 7/8. We think these data are enough to address reviewer's concern here. As reviewer suggested, we added an experiment plan and future directions in discussion section in our revised manuscript.

Further major issues:

Comments: 1) The validation of AS events has been extended and included as Figure 7. Most of the data look convincing, but the following modifications should be made. First, the legends says "...gene expression levels were analyzed...". However, this method does not allow quantifying gene expression as normalization to a reference transcript is missing and as the method is not quantitative. Only AS ratios can be deduced from this data.

Response: This is similar to the first major concerns raised by reviewer 3. Please see our above response.

Comments: The last two candidates are not meaningful from my point of view. Glyma.19G081300 gives only one band. If I get the model right, exon2 can be skipped or excluded. The reverse primer cannot bind to the variant lacking exon2, therefore co-amplification for ratio analysis cannot be used. Instead, qPCR for analyzing the level of the variant with exon2 relative to total expression (or to the exon3 variant, in case these are mutually exclusive exons) of this gene would be needed to support a switch in splicing variant ratios. Comments: Glyma.16G073600 gives only one band, and is included as a control that AS efficiency is not altered. How can this serve as a control for AS if no alternative splicing is visible in any sample? Has AS for this gene been reported before, or why has it been chosen?

Response: These are good suggestions. (1) Glyma.19G081300 is an example to demonstrate exon skipping in our assay, but it is confusing indeed. This is mainly because the exon we examined is very short. In the revised manuscript, we removed Glyma.19G081300 data but added co-amplification of two other genes Glyma.05G230000 and Glyma.19G108300 to support exon skipping events. These primers can amplify both spliced and unspliced bands, please see Fig. 7 in the revised manuscript. According to our RNA-seq data, Glyma.16G073600 cannot generate alternative splicing isoforms, hence we removed Glyma.16G073600 in the revised manuscript.

Comments: Statements in the text such as “....total level from two isoforms remains similar” (l. 418-419) and “...does not yield reduction of intron-spliced signals...” (l. 422) lack experimental support, as this method cannot be used to quantify single transcripts or total expression level.

Response: See first response to reviewer 3 above.

Comments: 2) Figure 6 contains a summary of the RNA-seq data with respect to the alternative splicing changes. Fig. 6a gives a summary of genes showing significant AS changes in the different samples. Separate Venn diagrams have been used for the four basic AS types. It would be more logical to display events here, and maybe provide corresponding gene numbers in parenthesis, for the following reason. One gene might have for example an IR event in one sample, and an ES event in another sample. There is no overlap in the AS event category, but it would give an overlap for the gene. I am not sure how the authors have dealt with this kind of candidates.

Response: Thank you for pointing this out. We display the events and provide corresponding gene numbers in parenthesis, please see Fig. 6 in the revised manuscript.

Comments: Furthermore, I am not sure if I interpret the data display in the Circos diagram correctly. What does “heat map view of gene density including differential alternative splicing...” (from corresponding legend) mean? Does every colored line indicate the position of a gene with an AS event, and does the line color somehow indicate how close genes with AS events are? Finally, the red lines are supposed to connect the position of the GmSKRP1 genes with genes containing SKRP1/Avr3c-regulated events. What can be concluded from this? I think it would be sufficient to indicate the positions of the AS events, genes, and GC content,

and move this diagram to Supplement, as it is purely descriptive.

Response: We agreed with reviewer's suggestion. In revised manuscript, we reorganized this figure and added some details in the legend. We moved the diagram to Supplementary fig. 16d.

Comments: 3) For several figures, it remains unclear what kind of replicates have been used. For example, legend to Fig. 4c states "Three independent experiments showed similar results". Legend to Fig. 4d says "Statistics of (c). Means and standard errors from replicates are shown..." and in the diagram it says $n = 10$. If I understand it correctly, three independent experiments have been performed, and 10 replicates from one experiment have been analyzed.

Response: Yes, the experiments were repeated for three times with similar results, and means and standard errors from ten replicates are shown. We corrected these in the revised manuscript.

Comments: Another example is legend to SFig. 18: "Experiments were repeated three times with similar results" In the graphs, it says $n = 3$. Are these technical replicates or biological replicates? Thus, a clearer and more detailed description would be needed in several legends.

Response: The experiment was replicated three times, means and standard errors from three biological replicates are shown, we corrected these in revised manuscript.

Comments: Some legends still do not contain any information on number of replicates and types of error bars, e.g. SFig. 1a, c, d.

Response: We corrected these in the revised manuscript.

Comments: 4) It is difficult to see the details of the gene models displayed in Fig. 8, in particular arrows designating primers and the symbols for premature termination are too small, and colors of arrows are indistinguishable on a print out. The display could be reduced to the relevant part of the gene model to increase size of the relevant regions. Furthermore, the display of the qRT-PCR data includes mutants of GmSKRP1 (mut) and PsAvr3c (M4). These data are not mentioned in the text. Based on my expectation that the wild type but not the mutant versions alter splicing efficiency, I think that the labels for "PsAvr3c" and "GmSKRP1mut" are swapped. The values are defined as "...splicing efficiencies (the ratio spliced RNA over unspliced..." (l. 440-441). I would expect that splicing efficiency represents the ratio $\text{spliced} / (\text{spliced} + \text{unspliced})$. Given that some values are clearly above 1, and as 1 would be the maximum, i.e. 100% spliced, I guess that the values were indeed calculated differently. The quantification would require a better description.

Response: We increase the size of gene models and added description of qRT-PCR data (includes mutants of GmSKRP1^{mut} and PsAvr3c^{M4}) in the revised manuscript. We realized that the label "PsAvr3c" and "GmSKRP1^{mut}" are swapped by mistake in Fig. 8. We corrected this in the revised manuscript. For the splicing efficiencies, the ratio represented here is $\text{spliced} / \text{unspliced}$, not $\text{spliced} / (\text{spliced} + \text{unspliced})$. This calculation is widely used as standard splicing efficiency validation in many previous publications (Wu Z, et al. The Plant cell 28, 55-73 (2016), Marquardt S, et al. Molecular cell 54, 156-165 (2014), etc.). We also added description in the figure legend and Materials and Methods part to clarify the calculation method.

Comments: 5) The authors express two of the splicing targets, Glyma.02G222300 and Glyma.03G220800, in hairy roots, which results in increased resistance to *P. sojae* (SFig. 17c-e). It is unclear what conclusion can be drawn from this experiment in the context of the splicing phenotype. First, are the genes overexpressed in the hairy roots? Transcript levels should be analyzed compared to the GFP control line. The authors show in their other experiments that the splicing change does not involve changes in total transcript levels. So how could the phenotype relate to changes in alternative splicing?

Response: The goal of this experiment is to examine the function of the two genes (WRKY and NAC transcriptional factor genes) in soybean immunity against *P. sojae* to address the concerns from reviewer 1 (functionally validate the role of NAC-TF and/or WRKY transcription factors in immunity to *P. sojae*). Therefore, we overexpressed two genes in soybean hairy root with cDNA sequences (no splicing is required).

To confirm these two genes overexpressed in the hairy roots, we complemented with qRT-PCR results. Please see SFig. 17 (d,e) in the revised manuscript. From the phenotype, we confirm these genes positively contribute to soybean defense. Because the AS isoforms of both NAC and WRKY contain premature stop codon, we therefore speculate that AS change by effector will attenuate functions of these immune related genes.

Second, were genomic or cDNA sequences used for these constructs, and can alternative splicing still occur?

Response: We expressed Glyma.02G222300 and Glyma.03G220800 cDNA in soybean hairy roots, and alternative splicing cannot occur.

Comments: 6) The manuscript still contains many grammatical errors. Usage of the terms mRNA/pre-mRNA is still not correct at several places, e.g. l. 75 "...over 60% of mRNA transcripts ...undergo AS". Several statements in introduction need clarification: 1. l. 70-71 "...mutually exclusive exons observations show that IR is a frequent event in both in human and in plants." - meaning unclear.

Response: We checked mRNA/pre-mRNA term throughout the manuscript, examine the grammar carefully in revised manuscript.

Comments:l. 81 "...many non-snRNPs are also involved in spliceosome operations." - Which ones?

Response: We modified this statement and cited publication in the revised manuscript.

Comments:3. l. 124 "...GmSKRPs, novel components of plant spliceosome." - referring to these proteins as interactors of spliceosomal components might be more appropriate as long as their molecular function is unclear.

Response: We agree that this statement is indeed more appropriate. We corrected this in revised manuscript.

Further comments:

Comments: 1) Legends: Most legends summarize the findings, sometimes even twice; e.g. legend to Fig. 2c: "PsAvr3c interacts with GmSKRP1/2 ... confirms that FLAG-PsAvr3c

specifically interacts with GFP-GmSKRP1/2..."; or legend to Fig. 2d: "Measuring binding affinity between PsAvr3c and GmSKRP1 ... binding affinity of PsAvr3c to GmSKRP1 was determined...". The legends could be reduced to a description of the figures, but should also sometimes provide more details.

Response: Thanks. We reduced those redundant statements in figure legend and carefully checked throughout the revised manuscript.

For example, for none of the immunoblots it is mentioned that labels with numbers represent molecular weight in kDa. This kind of information should be indicated in the figure or legend.

Response: We added the kDa in figures of the revised manuscript.

Comments: 2) Legend to SFig. 11: legend title incomplete; change to "...in soybean immunity against *P. sojae*"

Response: We corrected this in the revised manuscript.

Comments: 3) Legend to Supplemental Fig. 18: "... transcriptional level of NAC..." should be replaced by "... transcript level of NAC..." as these qPCR analyses measure steady state levels, i.e. sum of transcription and decay, and not transcription itself.

Response: Thank you for pointing this out, we corrected this in the revised manuscript.

Reviewers' comments:

Reviewer #1 (Remarks to the Author):

Having read the MS, i am satisfied with the authors' responses to my comments (many thanks).

Although i am not an expert in alternative splicing, i believe the data to be convincing and sufficient to demonstrate a role for Avr3c towards host AS. Clearly, many interesting avenues for follow up will emerge from this work from the authors as well as other groups in the plant-microbe and AS fields. Some minor comments are listed below:

Corrected text. Quality of some new texts and edits need to be improved. Some examples are provided below:

" However, we found many other quantitative co-amplification cases that do not fit NMD inhibition speculation". change speculation to scenario.

"To access the global genes that are subject to AS during infection by PsAvr3c effector, an alternative approach would be direct comparison of RNA-seq data from soybean tissues that are infected by wild type strain and PsAvr3c knockout mutant".

it is not clear to me what is meant here. identify genes that are subject to AS on the whole genome level?

"Validation of alternative splicing ratios of each gene were analyzed in hairy roots ectopically expressing different recombinant proteins (left panels) or wild-type roots infected by different strains of *P. sojae* (right panels) by RT-PCR with primers specific to each selected gene (Supplementary Table 1)".

it says that the validation of AS was analysed. this does not make sense to me. sentence needs to be revised to clarify (simplify).

Figure 8: Gene labels refer to Transcriptional factors. this should either be "transcription factors" or "transcriptional regulators"

Reviewer #3 (Remarks to the Author):

Huang et al. have provided another revised version of their manuscript. The manuscript has significantly improved by changing data presentation and organization, removing/exchanging data, correcting errors in figure labels and providing more details in the legends. The authors now provide also a discussion of the possibility that some AS event changes may be due to changes in NMD, as this cannot be excluded from their current data. However, their argumentation with respect to their AS analysis is still incorrect, and this is the only point where I see few further data needed to put their conclusion on solid ground. The authors argue that the reciprocal change in two bands from a co-amplification PCR indicates that

one has increased, while the other has decreased. However, due to intrinsic features of non-quantitative PCR, the PCR reaction will run at some point into limitation. So in case one splicing variant would be strongly overrepresented, e.g. due to inhibition of NMD turnover, co-amplification PCR would not only give a stronger band for this product, but also result in a weaker band for the alternative product, as PCR runs into limitation. This is a well-known phenomenon and other experts in the splicing field can be contacted for further advice. It would be sufficient if the authors show for few (e.g. three) examples that there are indeed reciprocal changes in the splicing variant levels, by performing RT-qPCR of individual splicing variants. The authors have included such an analysis for one candidate in the rebuttal letter. However, it is essential to include these experiments in the manuscript.

Further points:

- 1) The authors have now explained that they defined splicing efficiency as ratio of spliced to unspliced. While this kind of calculation is often used and fine, it should be referred to as splicing ratio and not splicing efficiency (which would correspond to spliced/total) throughout the text. Furthermore, from the coverage plot, no intron retention can be seen for the oxidoreductase control. So was the intron retained form detectable at all? If not, how can the splicing ratio shown in Fig. 8c be calculated? Overall, this control is not very informative, as it is well known that intron retention affects many but not all introns. A better control here would be an intron retention event, that is not affected by GmSKRP1.
- 2) l. 464-465: please rephrase, difficult to understand.
- 3) l. 471-472: text needs to be rephrased. Furthermore, it is difficult to conclude from total transcript levels (I assume this is what is displayed in SFig. 18) to levels of individual transcripts. For example, a change in a low abundant isoform may not result in a significant change in total transcript level. Would it not be surprising if a major shift between two splicing variants, of which one may be retained in the nucleus and the other translated in the cytosol, does not give a change in total transcript level? This would require identical stabilities of two transcript variants with very different fates. Furthermore, other regulatory mechanisms may mask changes in total transcript levels.
- 4) l. 563: nonsense-mediated decay, not nonesense mediated decay
- 5) l. 565-569: The interpretation of co-amplification experiments need to be corrected. Furthermore, the authors argue that the changes in AS events other than intron retention argue against an NMD involvement. This is not correct, as the other AS types also often give rise to NMD target features. Have the authors checked for the presence of NMD-triggering features in the splicing variants from those other events? Furthermore, for transcripts retained in the nucleus, turnover processes other than NMD may be involved. So the authors should have a more balanced discussion that the changes of splicing variant ratios can result from AS or altered decay. AS events, for which the authors include RT-qPCR data of individual splicing variants and find reciprocal changes, a stronger statement regarding the involvement of AS can be made.
- 6) l. 623-625. The text section is misleading, as RZ-1B/C are not SR proteins. Please clarify.

We thank the reviewers for constructive comments and giving us an opportunity to submit a revised manuscript. In the revised manuscript, we added new data to address the reviewers' concerns and incorporated their suggestions. Our point-by-point response to reviewers' concerns/comments is provided below.

Reviewer #1 (Remarks to the Author):

Having read the MS, i am satisfied with the authors' responses to my comments (many thanks). Although i am not an expert in alternative splicing, i believe the data to be convincing and sufficient to demonstrate a role for Avr3c towards host AS. Clearly, many interesting avenues for follow up will emerge from this work from the authors as well as other groups in the plant-microbe and AS fields. Some minor comments are listed below:

Comments: Corrected text. Quality of some new texts and edits need to be improved. Some examples are provided below: " However, we found many other quantitative co-amplification cases that do not fit NMD inhibition speculation". change speculation to scenario.

Response: We corrected the text in the revised manuscript.

Comments: "To access the global genes that are subject to AS during infection by PsAvr3c effector, an alternative approach would be direct comparison of RNA-seq data from soybean tissues that are infected by wild type strain and PsAvr3c knockout mutant". it is not clear to me what is meant here. identify genes that are subject to AS on the whole genome level?

Response: We corrected the text into "To assess the genes that are subjected to AS on the genome-wide level during infection by PsAvr3c effector....." in line 603.

Comments: "Validation of alternative splicing ratios of each gene were analyzed in hairy roots ectopically expressing different recombinant proteins (left panels) or wild-type roots infected by different strains of P. sojae (right panels) by RT-PCR with primers specific to each selected gene (Supplementary Table 1)".it says that the validation of AS was analysed. this does not make sense to me. sentence needs to be revised to clarify (simplify).

Response: We rephrased the sentences in the revised manuscript in lines 1343-1354.

Comments: Figure 8: Gene labels refer to Transcriptional factors. this should either be "transcription factors" or "transcriptional regulators"

Response: We corrected "transcriptional factor" into "transcription factor" throughout the revised manuscript.

Reviewer #3 (Remarks to the Author):

Comments: Huang et al. have provided another revised version of their manuscript. The manuscript has significantly improved by changing data presentation and organization, removing/exchanging data, correcting errors in figure labels and providing more details in the legends. The authors now provide also a discussion of the possibility that some AS event changes may be due to changes in NMD, as this cannot be excluded from their current data. However, their argumentation with respect to their AS analysis is still incorrect, and this is the

only point where I see few further data needed to put their conclusion on solid ground. The authors argue that the reciprocal change in two bands from a co-amplification PCR indicates that one has increased, while the other has decreased. However, due to intrinsic features of non-quantitative PCR, the PCR reaction will run at some point into limitation. So in case one splicing variant would be strongly overrepresented, e.g. due to inhibition of NMD turnover, co-amplification PCR would not only give a stronger band for this product, but also result in a weaker band for the alternative product, as PCR runs into limitation. This is a well-known phenomenon and other experts in the splicing field can be contacted for further advice. It would be sufficient if the authors show for few (e.g. three) examples that there are indeed reciprocal changes in the splicing variant levels, by performing RT-qPCR of individual splicing variants. The authors have included such an analysis for one candidate in the rebuttal letter. However, it is essential to include these experiments in the manuscript.

Response: We appreciate the reviewer's comments that providing qPCR data of each splicing variant would be more accurate (We've learnt a lot from a series of reviewer 3's comments, many thanks). We agree that co-amplification has limitations in validating AS events. As suggested by this reviewer, we performed RT-qPCR to measure transcript level of each splice variant from three selected soybean genes. We added this data in the revised manuscript (see SFig. 17).

We agree that differential turnover of splice isoforms by NMD could also be responsible for the observed changes in splice isoforms. However, due to technical issues in soybean transformation, it may be difficult to examine inhibition of NMD by genetic approaches. Hence, we discussed this possibility in the revised manuscript and leave this topic open for future research. We hope that our current data could support our statement on AS.

Further points:

Comments: 1) The authors have now explained that they defined splicing efficiency as ratio of spliced to unspliced. While this kind of calculation is often used and fine, it should be referred to as splicing ratio and not splicing efficiency (which would correspond to spliced/total) throughout the text. Furthermore, from the coverage plot, no intron retention can be seen for the oxidoreductase control. So was the intron retained form detectable at all? If not, how can the splicing ratio shown in Fig. 8c be calculated? Overall, this control is not very informative, as it is well known that intron retention affects many but not all introns. A better control here would be an intron retention event, that is not affected by GmSKRP1.

Response: Thank you for pointing this out. Regarding the way we calculated AS in our manuscript, we corrected the "splicing efficiency" to "splicing ratio" throughout the text.

We agree that a proper control is important here. In Fig. 8c of the revised manuscript, we updated a soybean COP9 signalosome complex subunit gene Glyma.03G016800 as a control. Glyma.03G016800 was previously named as Glyma03g01880 and had an intron retention event (Shen et al. Plant cell, 2014). Here, the alternative splicing is not affected by GmSKRP1 and PsAvr3c.

Comments: 2) l. 464-465: please rephrase, difficult to understand.

Response: We rephrased this part in the revised manuscript.

Comments: 3) l. 471-472: text needs to be rephrased. Furthermore, it is difficult to conclude from total transcript levels (I assume this is what is displayed in SFig. 18) to levels of individual transcripts. For example, a change in a low abundant isoform may not result in a significant change in total transcript level. Would it not be surprising if a major shift between two splicing variants, of which one may be retained in the nucleus and the other translated in the cytosol, does not give a change in total transcript level? This would require identical stabilities of two transcript variants with very different fates. Furthermore, other regulatory mechanisms may mask changes in total transcript levels.

Response: Thank you for pointing this out. We realized that it is not suitable to discuss these possibilities in result part. Therefore, we remove speculation and soften the statement in this paragraph.

Comments: 4) l. 563: nonsense-mediated decay, not nonesense mediated decay

Response: We corrected the text in the revised manuscript.

Comments: 5) l. 565-569: The interpretation of co-amplification experiments need to be corrected.

Response: We rewrite this section in the revised manuscript.

Comments: Furthermore, the authors argue that the changes in AS events other than intron retention argue against an NMD involvement. This is not correct, as the other AS types also often give rise to NMD target features. Have the authors checked for the presence of NMD-triggering features in the splicing variants from those other events? Furthermore, for transcripts retained in the nucleus, turnover processes other than NMD may be involved. So the authors should have a more balanced discussion that the changes of splicing variant ratios can result from AS or altered decay.

Response: We agreed that NMD could also contribute to the observed changes in splice isoforms. We therefore revised the related paragraph and added more balanced discussion in the revised manuscript.

Comments: AS events, for which the authors include RT-qPCR data of individual splicing variants and find reciprocal changes, a stronger statement regarding the involvement of AS can be made.

Response: Similar to the last two comments, we rewrote the AS discussion part in our revised manuscript.

Comments: 6) l. 623-625. The text section is misleading, as RZ-1B/C are not SR proteins. Please clarify.

Response: We reworded the sentence in the revised manuscript in lines 631-633.

REVIEWERS' COMMENTS:

Reviewer #3 (Remarks to the Author):

In their latest revised version, the authors have adequately addressed all of my concerns. I'm looking forward to seeing this work published.